# Learning a Data-Driven Policy Network for Pre-Training Automated Feature Engineering

**Liyao Li**[1]     **Haobo Wang**[1]     **Liangyu Zha**[2]     **Qingyi Huang**[2]
**Sai Wu**[1]     **Gang Chen**[1]     **Junbo Zhao**[1][*]
[1]College of Computer Science and Technology, Zhejiang University
[2]Institute of Computing Innovation, Zhejiang University

## Abstract

Feature engineering is widely acknowledged to be pivotal in tabular data analysis and prediction. Automated feature engineering (AutoFE) emerged to automate this process managed by experienced data scientists and engineers conventionally. In this area, most — if not all — prior work adopted an identical framework from the neural architecture search (NAS) method. While feasible, we posit that the NAS framework very much contradicts the way how human experts cope with the data since the inherent Markov decision process (MDP) setup differs. We point out that its data-unobserved setup consequentially results in incapability to generalize across different datasets as well as also high computational cost. This paper proposes a novel AutoFE framework *Feature Set Data-Driven Search* (Fetch[1]), a pipeline mainly for feature generation and selection. Notably, Fetch is built on a brand-new data-driven MDP setup using the tabular dataset as the *state* fed into the policy network. Further, we posit that the crucial merit of Fetch is its *transferability* where the yielded policy network trained on a variety of datasets is indeed capable to enact feature engineering on unseen data, without requiring additional exploration. This is a pioneer attempt to build a tabular data pre-training paradigm via AutoFE. Extensive experiments show that Fetch systematically surpasses the current state-of-the-art AutoFE methods and validates the transferability of AutoFE pre-training.

## 1 Introduction

*Tabular data* — also known as *structured data* — abound in the extensive application of database management systems. Modeling tabular data with machine learning (ML) models has greatly influenced numerous domains, such as advertising (Evans, 2009), business intelligence (Quamar et al., 2020; Zhang et al., 2020), risk management (Babaev et al., 2019), drug analysis (Vamathevan et al., 2019), etc. In resemblance to the other data forms like images or text, building a proper *representation* for the tabular data is crucial for guaranteeing a decent system-wide performance. In this regime, this process is also known as *feature engineering* (FE), which was conventionally conducted by highly experienced human experts. In other words, as many empirical studies show (Heaton, 2016), FE almost always serves as a necessary prerequisite step in ML modeling pipelines.

The recent advances in reinforcement learning (RL) have provided a new possibility for automated feature engineering (AutoFE) and automated machine learning (AutoML). Neural architecture search (NAS) (Zoph & Le, 2016) has nearly become a synonym for AutoML in the field of computer vision, based on an RL setup dedicated to searching for undesigned neural network architectures with excellent performance. As for tabular data, a series of well-known open-source packages (such as TPOT (Olson & Moore, 2016), AutoSklearn (Feurer et al., 2015) and AutoGluon (Erickson et al., 2020)) claim to implement the AutoML pipeline. However, they do not generally cover AutoFE, especially feature construction and selection, which is supposed to be part of AutoML as shown in Figure 1. To date, AutoFE has been a significant and non-negligible compo-

---

[*]Correspondence to `j.zhao@zju.edu.cn`.
[1]Source code is available at `https://github.com/liyaooi/FETCH`, implemented by Mindspore.

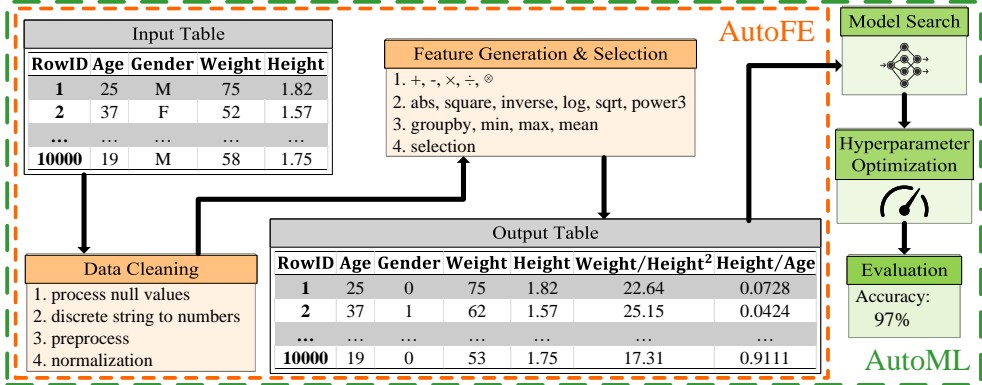

Figure 1: The different pipelines of automated feature engineering (AutoFE) and automated machine learning (AutoML). Existing AutoML frameworks focus more on model setup and lack concern for feature engineering.

nent of AutoML on tabular datasets because it not only constructs better features to facilitate model fitting but also enjoys high interpretability.

Recently, the mainstream line of AutoFE is based on a *trial-and-error* NAS-like process. For instance, Neural feature search (NFS) (Chen et al., 2019) was introduced to find the best performing FE plans by drawing inspiration from the NAS framework — for the first time embedding *past FE actions* into its policy network as the *states* in its Markov decision process (MDP) for RL training to iteratively select better FE actions. The follow-up work DIFER (Zhu et al., 2022) extends its NAS-like setup to a differentiable one. However, a data scientist or engineer usually tends to investigate the data — such as analyzing its distribution, identifying the outliers, measuring the correlation between columns, etc. — and then proposes an FE plan. They may further use the derived plan to test the prediction performance and repeat this process considering the evaluated score. Meanwhile, they also can accumulate knowledge to accelerate decision-making.

As we scrutinize these works, we posit that existing NAS-like AutoFE frameworks on tabular data have two shortcomings, largely deviating from how human experts cope with the data. First, they have stuck themselves with the **data-unobserved** paradigm because their policy network does not even see the tabular data itself and proposes data-unrelated FE plans. Second, the inherent data-unobserved setup makes them **lack transferability**, unfeasible to borrow knowledge from previous training experience to speed up the exploration process when facing a completely new dataset.

This paper hopes to bridge this methodology gap between the human experts and data-unobserved methods for AutoFE and validate its feasibility based on the above discussions. In particular, we establish a new form of MDP setup where the *state* is defined simply as a processed dataset drawn from its original counterpart. The policy network yielded is a succinct mapping from the input data table directly to its (sub)optimal feature engineering actions plan. To this end, we present **FE**ature **SET** DaTa-Driven Sear**CH** (FETCH) — a brand new RL-based framework for AutoFE but with a completely distinct *data-driven* MDP setup to emulate the human experts. As shown in Figure 2, FETCH outputs FE actions

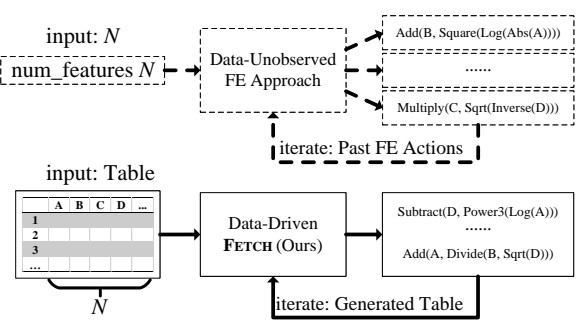

Figure 2: The difference between data-driven FETCH and data-unobserved approach. See text for details.

well-designed for the input data, and iteratively constructs more appropriate actions based on the newly generated data. In contrast, traditional data-unobserved methods only take in the number of features to be processed and iteratively update with the sequence of past actions.

Thanks to the aforementioned design principles of FETCH, another favored by-product is that it enables *transferability* by pre-training for the AutoFE workflow. Simply put, we validate that FETCH can be pre-trained in a collaborative manner where we feed multiple tabular datasets and maximize

the expected sum of the reward altogether. In particular, we could directly exploit it to unseen datasets, with little or even no further exploration required. The "transferability" underlines that the policy network in FETCH manages to accumulate knowledge from the fed datasets to be able to transfer them to other scenarios. We argue that this behavior of FETCH is indeed similar to the human experts' in that the richer experiences accumulated from handling various types of datasets may offer them better insights when facing the unseen ones. It is also worth noting that the NAS-like AutoFE frameworks would demand exploration from scratch given any new dataset, profoundly due to their inherent inability to derive associations across datasets with their data-unobserved MDP. Last but not the least, this can also be understood from a standpoint of *pre-training* — connected to what has been recently prompted in other domains — such as BERT (Devlin et al., 2018) / GPT-3 (Brown et al., 2020) for natural language processing or the SimCLR (Chen et al., 2020) for computer vision.

To sum up, the main contributions are listed as follows.

1. We identify a crucial methodology gap between the data scientist/engineer with the current existing AutoFE frameworks.
2. We propose a novel AutoFE framework for both classification and regression tasks, dubbed FETCH, on bridging this gap, including a data-driven feature engineering pipeline for data cleaning, feature generation and selection. Empirical results show its on-par or superior performances to the previous state-of-the-art method.
3. For the first time, we characterize a *transferability* principle for AutoFE. It reflects how much knowledge or experience a trained policy may be able to accumulate to enable the exploration of unseen datasets. FETCH concept-proves its feasibility. This is also linked to the pre-training paradigm.

## 2   RELATED WORK

**Automated feature engineering**   AutoFE aims to find the best-performing set of features on feature engineering plans (e.g. transformation or selection) with minimal human intervention. For instance, the LFE algorithm (Nargesian et al., 2017) pioneers parametrizing the feature transformation that is only applicable to classification problems. To generate higher-order features, Trans-Graph (Khurana et al., 2018) was introduced by formulating the feature engineering plan via a data structure of the graph and Q-learning (Watkins & Dayan, 1992) algorithm. While conceptually promising, this approach is shown to suffer from severe feature dimension explosion problems where excessive feature columns are appended to the raw data. The Neural feature search (NFS) framework (Chen et al., 2019), primarily stealing the setup from the RL-based NAS architecture (Zoph et al., 2018), shed some light on how to tackle this problem. It employs a separate recurrent neural network (RNN) controller (Medsker & Jain, 2001) to generate a new feature for every feature column individually, yielding that N features require N computationally expensive RNN controllers. On a separate but related line, the differentiable NAS framework (Luo et al., 2018) is employed to perform AutoFE, named DIFER (Zhu et al., 2022) by transferring a discrete optimization problem to be continuous. However, NFS and DIFER only use the data for feature evaluation, not for feature generation. Also, they only evaluate the effect of each individual feature, rather than considering the comprehensive performance of the entire generated feature set. While this data-unobserved approach to constructing features works to some extent, we believe it is inconsistent with the pattern of human experts scrutinizing the data before proposing feature engineering plans.

## 3   OVERVIEW OF FETCH

### 3.1   THE FE CONTROL PROBLEM

Given a prediction-involved problem with its dataset $\mathbf{D} = (\mathbf{X}, \mathbf{Y})$ containing: (i)-a set of features, $\mathbf{X} = \{x_1, x_2 \dots x_d\}$ where $\mathbf{X} \in \mathbb{R}^{n \times d}$ denotes the tabular data containing $n$ rows (instances) and $d$ columns (features); (ii)-a corresponding target label vector, $\mathbf{Y}$, which can be either discrete or continuous, compatible with classification or regression problems respectively. Similar to most prior work around AutoFE (Khurana et al., 2018; Chen et al., 2019; Zhu et al., 2022), a pre-selected learning algorithm $\mathbf{L}$ (e.g. Random Forest Classifier or Xgboost) with its hyperparameters fixed

and a measure of cross-validation performance $\mathbf{E}$ (e.g. F1-score) are considered in the experimental verification.

In addition, we denote $\mathcal{T} = \{t_1, t_2 \ldots t_m\}$ as a feature engineering plan where it consists of an ordered sequence of $m$ feature transformation actions $t$ initiated from the raw data. Each transformation action $t$ in $\mathcal{T}$ is applied on an internally specified feature $x$ instead of whole features $\mathbf{X}$. The whole set of derived features that may be deduced from $\mathbf{X}$ through $\mathcal{T}$ is referred to as $\hat{\mathbf{X}}_{\mathcal{T}}$. Note that, transformations can lead to a higher-order feature (i.e. combined by three or more features) because it is equivalent to applying binary operation recursively.

The purpose (or objective function) of feature engineering is defined as Equation 1. Given a dataset $\mathbf{D} = (\mathbf{X}, \mathbf{Y})$ with a set of original features $\mathbf{X}$ and the target $\mathbf{Y}$, search a sequence of transformation actions $\mathcal{T}$ to derive a transformed feature set $\hat{\mathbf{X}}_{\mathcal{T}}$, which maximizes the cross-validation performance $\mathbf{E}(\mathbf{L}(\hat{\mathbf{X}}_{\mathcal{T}}, \mathbf{Y}))$ for a given algorithm $\mathbf{L}$ and a metric $\mathbf{E}$.

$$\mathcal{T} = \arg\max_{\mathcal{T}} \mathbf{E}(\mathbf{L}(\hat{\mathbf{X}}_{\mathcal{T}}, \mathbf{Y})) \tag{1}$$

## 3.2 RL SETTING OF FETCH

The RL learning scheme, as illustrated in Figure 3, is employed in order to output an extended tabular dataset concatenated with generated and selected features. The core to FETCH is a policy network of the RL agent which is often instantiated by a well-designed deep neural network model introduced in the following Section 3.3. The policy network takes the dataset itself, or a bootstrapped counterpart to cope with the scaled dataset as a *state*, and produces a feature engineering plan through an inference pass. **In that regard, quite different from prior work of AutoFE, FETCH is — in theory — capable to fully observe, perceive, and recognize the pattern concealed in the dataset.** At its core, FETCH formulates feature engineering as a **control** problem, elaborated in above Section 3.1. Diving into this control problem, we propose a complete MDP setup including an *environment* involving an observed dataset and a *reward* function assessing the quality of the FE plans. The environment allows the agent to first employ an FE actions/operations sequence to generate a new table, then receive back a positive/negative reward vector that summarizes its results, derived from the reward function. The goal of the RL agent is to learn, by repeated interactions with the environment, a policy network capable of mapping input data to a sequence of FE actions with maximum cumulative reward. Once the learning phase is finished, the output tabular data provided to the downstream ML model is generated upon the sequence of FE actions. To enable transferability to a certain degree in FETCH, we facilitate the training phase of this control problem via the reinforcement learning (RL) paradigm. Due to space limitation, the MDP detail settings such as FE action space and reward function $\mathcal{R}$ are covered in Appendix A.1.

As the informative architecture of FETCH shown in Figure 3, FETCH outputs a sequence of transformation actions plan $\mathcal{T} = \{t_1, t_2 \ldots t_m\}$ after a tabular dataset is conceived by our framework. To get the final prediction result, we sequentially apply the actions list on the tabular data $\mathbf{X}$ and then feed it into an ML predictor (e.g. LR, RF, Xgboost) for the feature set evaluation and reward calculation. This process is identically employed in both the training and evaluation phases. Within each rollout, at time step $i$, we run the current data $\mathbf{X}_i$ through the policy network $\pi$ to get an action probability over pe-supported operators. We then sample the sequence of transformation actions $\mathcal{T}$ following multinomial distribution by softmax. For the higher-order feature generation search, while we only include binary operators in the set, we generally run FETCH by multiple steps so that it covers more complex higher-order features that may involve more than two feature vectors. We let $\mathcal{T}$ interact with the environment yielding transformed tabular data $\mathbf{X}_{i+1}$ which integrates the input $\mathbf{X}_i$. We repeat this process multiple times until it reaches a pre-defined limit value $K$ or converges.

A table generated by FETCH on a healthcare example is depicted in Figure 1. As shown in the figure, FETCH initially cleans and encodes the data so that the label *M/F* in the column *Gender* becomes *0/1* respectively. Then the processed data is fed into our policy network, which learns how to generate and select features, and eventually applies appropriate FE actions to the input table to obtain the output table. For improved readability, each generated feature is named by the action path that constructs it. For example, a more effective feature *Weight/Height*$^2$ (a.k.a Body Mass Index (Lee et al., 2013)) is calculated by column *Weight* and the square of column *Height*. This interpretable path-like illustration of the operations allows experts to easily understand feature meanings and strengthens interpretability.

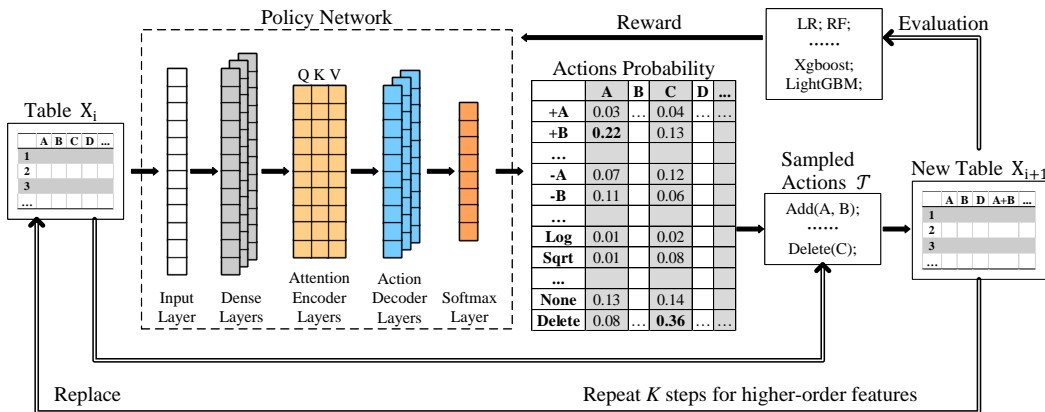

Figure 3: An architecture of the data-driven FETCH framework. For step $i$ in each epoch, the newly generated feature set $\mathbf{X}_{i+1}$ will replace its previous counterpart $\mathbf{X}_i$ to construct higher-order features and further selection. The subfigure of the middle dashed box reveals the micro-composition of modules inside the policy network.

## 3.3 POLICY NETWORK ON FEATURE SET

As discussed above, the RL agent in FETCH framework is primarily represented by a policy network $\pi$, where the main challenge of architectural design is centered on. Notably, to characterize the tabular dataset and properly learn a decent representation of it, we must let the policy network be capable of handling the *permutation-invariant* and *variable-length* natures of tabular data.

In contrast to the data forms of computer vision or natural language, when swapping or permuting the columns of tabular data, it remains the same dataset. This property is identical to a "set" data form as permutation invariant, which requires the neural network to be able to perceive the lists of different sequential features as still the same data set. When conducting feature engineering, the column number of the tabular would often change. Some of the column features might be replaced or deleted, while some others (especially for iterative generation of higher-order features) might get appended. This property of variable length demands that the neural network be more flexible with respect to the shape of the input data. Note that, the nature of this data form has prevented us from employing the convolutional network family or the recurrent network family because these models encode positions as part of their inductive bias.

To that end, we finalize our design by viewing the tabular data as a set of feature (column) vectors. Specifically, we draw inspiration from the Transformer architecture (Vaswani et al., 2017) and we carefully choose our neural operators to equip into the policy network. A brief depiction of it is shown in the middle (Policy Network) of Figure 3.

The primary components of our policy network are the following: (i)-a fully-connected network (FCN) containing an input layer and dense layers. It changes the dimension of a *variable-length* feature vector to a fixed length; (ii)-a multi-head self-attention module containing attention encoder layers (more details in Vaswani et al. (2017)). It measures the complex relationship between features in the set, which can be regarded as encoding the relationship of the feature columns; (iii)-an action output module including action decoder layers and softmax layer. It decodes the output of the upstream encoder to map the correlation information of the features to the corresponding action probabilities. Unlike transformers, we abandon the positional encoding from the multi-head self-attention module to satisfy the *permutation-invariance* property due to the nature of this data form. All these parametric modules allow arbitrary permutation of the feature vectors and can tackle the variable size of the feature set. Further, we reason that the stacked modules enable representation learning, coupled with the self-attention mechanisms effectively gauging the correlation and relationship between the feature columns in the set. The multi-layer setup also assists the higher-order feature discovery.

The feature-set-based action sampling rules are applied to the probability of the actions yielded by the policy network to propose an FE action plan (Appendix A.2). Proximal Policy Optimization (PPO) (Schulman et al., 2017) is adopted as the RL training manner in Algorithm 1 (Appendix A.3).

### 3.4 TRANSFERABILITY VIA PRE-TRAINING

With data-driven FETCH, we demonstrate that our approach very much resembles the human experts who can quickly transfer experience in handling unseen datasets, and it exhibits a certain degree of transferability. This setup also links to the recent popular topic of pre-training, such as BERT (Devlin et al., 2018) or SimCLR (Chen et al., 2020) in natural language processing and computer vision respectively. The goal of the pre-training paradigm is to drive the model to learn "common-sense" through large-scale training over a variety of data, with either supervised or unsupervised loss functionals. Common sense is manifested in the representation space, quite aligning the goal of feature engineering for tabular data. Despite the promise, the research of pre-training on the tabular data is very much unaddressed. With FETCH, we hope to provide the first viable technical solution and empirically validate its feasibility. The corresponding experiment of transferability can be found in Section 4.3.

Formally, given a set of datasets $\mathcal{D} = \{\mathbf{D}_1, \mathbf{D}_2 \ldots \mathbf{D}_{N_d}\}$ where $N_d$ denotes the number of datasets occurring in the set. We notice that FETCH does not require these tables to possess identical or similar schemas, nor identical row/column sizes. In particular, closely following the procedures displayed in Algorithm 1, we blend these datasets into a key-value data pool struct and initialize the RL rollouts. Correspondingly, the reward is eventually yielded by $\sum_1^{N_d} \mathcal{R}(\mathbf{D}_i)$, according to Equation 2 and line 13 in Algorithm 1. Post to training on this blended set of data tables, we serialize and save the parameter of the attention encoder module in the policy network. Noted, despite that in the normal training scenarios for pre-train models in the domains like vision or language understanding, the input data must conform to a pre-defined shape and form. FETCH gets rid of this limitation thanks to its architectural design explained in Section 3.3, to guarantee the capability for the various shapes of data. We believe this flexibility is quite essential for the real-world application of FETCH, because the tabular data is deemed business-related so it lacks standardized preprocessing pipelines like image reshaping or text truncating.

In what follows, a set of unseen data tables would be prepared for testing, $\mathcal{D}' = \{\mathbf{D}_1', \mathbf{D}_2' \ldots \mathbf{D}_{N_d'}'\}$ where $N_d'$ denotes the number of testing sets presented. In the corresponding experiment (Section 4.3), we simply run the yielded policy network through these datasets and assess the overall performance as per chosen evaluation metrics. **There have not been any AutoFE or AutoML workarounds for tabular data managing to accomplish this setup of transferability.**

We postulate that FETCH achieves this mainly by (i)-having the right MDP setup where the states are grounded by drawing from the original datasets, paving the way for mapping data to corresponding FE actions; (ii)-having the proper architectural design of the policy network where it treats the tabular data as a *set* and satisfies the appropriate properties (i.e. permutation-invariance and variant-length). One may ask, why could this work? In hindsight, with this scheme of pre-training on various datasets, FETCH is encouraged to (i)-reinforce the better feature engineering actions through vast exploration and (ii)-build a proper mapping relationship between the input data to the action space, with regard to a different form, distribution of the data table (or its columns), etc. The NAS-like methods only fulfill (i) but fail at (ii), which disallows any pre-training and transferability across unseen tables. The (ii) in FETCH very much accords to the human professionals because the experienced ones absorb experiences and are often capable to provide a preliminary feature engineering plan simply by scrutinizing the data — that may or may not require further trial-and-error processes. The corresponding experiment concept-proofing the transferability and generalizability of FETCH are provided in Section 4.3.

## 4 EXPERIMENTS

### 4.1 EXPERIMENTAL SETTING

**Goals of experiments**   FETCH is proposed to simulate two major characteristics of human experts in operating feature engineering processes: *data-driven observation* and *the transferability of exploration experience*. The first goal is to assess the superiority of our innovative data-driven MDP setup compared to other data-unobserved AutoFE methods in Section 4.2. The second goal is to verify the feasibility of the transferability brought by our setup in Section 4.3, i.e., whether it can converge faster or search for feature engineering action sequences with better performance by pre-training.

We adopt the same testing protocol (using the same evaluation model Random Forest) on identical benchmarks as most notable prior work, such as NFS and DIFER. Section 4.4 verifies that FETCH can boost many ML models. The ablation study in Appendix C presents the efficiency of FETCH and the influence of generated higher-order features. The fairness settings of experiments and the hyperparameters settings of various comparison methods are in Appendix B.2.

**Performance indicators**  For the classification tasks, the micro F1-score (the average harmonic mean of precision and recall) is used as the main metric and $(1 - (relative\ absolute\ error))$ (Shcherbakov et al., 2013) for the regression tasks. Both evaluation metrics are that the higher the score, the better the performance. Meanwhile, we also use the percentage of score improvement in the baseline score as a metric of improved efficacy. They are measured using 5-fold cross-validation.

**Datasets**  The experiments are conducted on 27 datasets including 11 regression (R) datasets and 16 classifications (C) datasets. These datasets are from OpenML[2], UCI repository[3], and Kaggle[4]. All datasets are available via the URL links in Appendix B.3. 5 datasets are further utilized to test the transferability by pre-training in Section 4.3. As a preprocessing step, we clean and encode the raw data using FETCH, ensuring that the data is readable by various ML models. And the final data input to each method is the same.

## 4.2 EFFECTIVENESS OF FETCH

**Comparison with AutoFE Methods**  We select six current widely-used or state-of-the-art **AutoFE** methods to show the effectiveness of FETCH on feature engineering. **Base** represents the baseline method evaluated on the raw dataset without any FE transformation. **Random** method randomly generates FE action sequences by Monte-Carlo exploration. Here we take the highest score among the same number of explorations as FETCH. **DFS** (Kanter & Veeramachaneni, 2015)[5] and **AutoFeat** (Horn et al., 2019) are famous and open-source Python libraries for search-based feature engineering. **NFS** (Chen et al., 2019) is the current best RL-based NAS-like AutoFE method, which employs the RNN module as controller of each original feature, and generates new features including higher-order ones by a series of actions for the original features. **DIFER** (Zhu et al., 2022) is the state-of-the-art AutoFE method, which optimizes the features by a differential NAS-like framework

As shown in Table 1, FETCH generally outperforms all the existing work by a significant margin. On benchmarking datasets, FETCH achieves state-of-the-art performance on 25 out of 27 datasets overall and gets a close second place in the remaining datasets. Although DIFER and NFS greatly outperform the baseline method, FETCH still performs 3.24% higher than DIFER and 3.16% higher than NFS on average, indicating the superiority of our data-driven setup in the AutoFE pipeline.

**Comparison with AutoML Methods**  To further highlight the superiority of FETCH, we additionally involve two famous **AutoML** methods. **AutoSklearn** (Feurer et al., 2015) is a popular open-source AutoML toolkit focusing on algorithm selection and hyperparameter tuning. **AutoGluon** (Erickson et al., 2020) is a full-fledged open-source AutoML framework developed by Amazon Inc., which covers many types of data. In particular, here uses the AutoGluon-Tabular of it.

Compared with the AutoML frameworks which work on model search and hyperparameter optimization, FETCH still has a great advantage, with 18 out of 27 datasets in total performing better than them. This serves as strong evidence that FETCH outperforms the other existing methods due to its human-expert-like data-driven setup with outstanding effectiveness. As a member of the AutoML family, these two baselines mostly focus on the model part and barely involve the feature engineering part. We believe these comparisons would further highlight the effectiveness of FETCH even beyond the scope of feature engineering.

---

[2] https://www.openml.org/

[3] https://archive.ics.uci.edu/ml/index.php

[4] https://www.kaggle.com/

[5] DFS is implemented in FeatureTools Toolkit ( https://featuretools.alteryx.com/en/stable/)

Table 1: Effectiveness comparison of FETCH with other AutoFE and AutoML methods. **Bold** indicates superior results amongst AutoFE methods. Note that AutoML methods focus on model search instead of feature engineering.

| Dataset | C/R | Instances\Features | AutoFE Methods | | | | | | | AutoML Methods | |
| --- | --- | --- | --- | --- | --- | --- | --- | --- | --- | --- | --- |
| | | | Base | Random | DFS | AutoFeat | NFS | DIFER | FETCH | AutoSklearn | AutoGluon |
| Airfoil | R | 1503\5 | 0.5068 | 0.6211 | 0.6003 | 0.5955 | 0.6226 | 0.6125 | **0.6463** | 0.5151 | 0.5083 |
| BikeShare DC | R | 10886\11 | 0.9880 | 0.9989 | 0.9990 | 0.9891 | 0.9991 | 0.9995 | **0.9997** | 0.9911 | 0.9967 |
| House King County | R | 21613\19 | 0.6843 | 0.6838 | 0.6908 | 0.6917 | 0.6934 | 0.6948 | **0.7475** | 0.7005 | 0.7442 |
| Housing Boston | R | 506\13 | 0.4641 | 0.4788 | 0.4708 | 0.4703 | 0.4977 | 0.5072 | **0.5224** | 0.4403 | 0.4857 |
| Openml_586 | R | 1000\25 | 0.6564 | 0.6646 | 0.7188 | 0.7178 | 0.7223 | 0.6946 | **0.7671** | 0.7297 | 0.7904 |
| Openml_589 | R | 1000\25 | 0.6395 | 0.6285 | 0.6956 | 0.7278 | 0.7165 | 0.6789 | **0.7562** | 0.7183 | 0.7998 |
| Openml_607 | R | 1000\50 | 0.6363 | 0.6392 | 0.6815 | 0.6499 | 0.6485 | 0.6564 | **0.7404** | 0.7265 | 0.7694 |
| Openml_616 | R | 500\50 | 0.5605 | 0.5834 | 0.5807 | 0.5927 | 0.5856 | 0.5982 | **0.6749** | 0.6618 | 0.6743 |
| Openml_618 | R | 1000\50 | 0.6351 | 0.6277 | 0.6848 | 0.6374 | 0.6461 | 0.6553 | **0.7351** | 0.7198 | 0.7520 |
| Openml_620 | R | 1000\25 | 0.6309 | 0.6288 | 0.6528 | 0.6574 | 0.6943 | 0.7262 | **0.7506** | 0.7199 | 0.7855 |
| Openml_637 | R | 500\50 | 0.5160 | 0.5478 | 0.5105 | 0.5763 | 0.5739 | 0.6006 | **0.6453** | 0.6416 | 0.6742 |
| Adult Income | C | 48842\14 | 0.8478 | 0.8485 | 0.8502 | 0.8483 | 0.8497 | **0.8584** | 0.8537 | 0.8629 | 0.8738 |
| Amazon Employee | C | 32769\9 | 0.9450 | 0.9442 | 0.9451 | 0.9453 | 0.9461 | 0.9474 | **0.9479** | 0.9471 | 0.9473 |
| Credit Default | C | 30000\25 | 0.8044 | 0.8089 | 0.8056 | 0.8086 | 0.8101 | 0.8108 | **0.8114** | 0.8194 | 0.8214 |
| Credit_a | C | 690\6 | 0.8362 | 0.8665 | 0.8216 | 0.8581 | 0.8695 | 0.8638 | **0.8754** | 0.8623 | 0.8377 |
| Fertility | C | 100\9 | 0.8700 | 0.8947 | 0.7900 | 0.8910 | **0.9189** | 0.8800 | 0.8900 | 0.8400 | 0.8800 |
| German Credit | C | 1001\24 | 0.7390 | 0.7738 | 0.7490 | 0.7600 | 0.7786 | 0.7730 | **0.7910** | 0.7460 | 0.7590 |
| Hepatitis | C | 155\12 | 0.8258 | 0.8639 | 0.8516 | 0.8677 | 0.8766 | 0.8839 | **0.9290** | 0.8065 | 0.7871 |
| Ionosphere | C | 351\34 | 0.9237 | 0.9514 | 0.9373 | 0.9286 | 0.9543 | 0.9515 | **0.9716** | 0.8194 | 0.8214 |
| Lymphography | C | 690\6 | 0.8315 | 0.8480 | 0.8113 | 0.8453 | 0.8614 | 0.8827 | **0.9260** | 0.8418 | 0.8522 |
| Megawatt1 | C | 4900\12 | 0.8655 | 0.8706 | 0.8813 | 0.8893 | 0.9167 | 0.9089 | **0.9209** | 0.8853 | 0.8850 |
| Messidor Features | C | 1150\19 | 0.6594 | 0.7026 | 0.7089 | 0.7359 | 0.7417 | 0.7541 | **0.7689** | 0.7402 | 0.7255 |
| PimaIndian | C | 768\8 | 0.7566 | 0.7609 | 0.7540 | 0.7643 | 0.7784 | 0.7839 | **0.7969** | 0.7462 | 0.7631 |
| SpamBase | C | 4601\57 | 0.9154 | 0.9211 | 0.9198 | 0.9237 | 0.9341 | 0.9372 | **0.9405** | 0.9272 | 0.9042 |
| SpectF | C | 267\44 | 0.7751 | 0.8221 | 0.8125 | 0.8331 | 0.8608 | 0.8538 | **0.8838** | 0.7828 | 0.7010 |
| Wine Quality Red | C | 999\12 | 0.5597 | 0.5774 | 0.5422 | 0.5641 | 0.5814 | 0.5779 | **0.6042** | 0.5804 | 0.5729 |
| Wine Quality White | C | 4900\12 | 0.4976 | 0.5046 | 0.4855 | 0.5023 | 0.5111 | 0.5153 | **0.5235** | 0.5376 | 0.5259 |

## 4.3 TRANSFERABILITY OF FETCH

As discussed in Section 3.4, FETCH is the first work that enables transferring "experience" across different datasets and realizes structured data pre-training. In this subsection, the transferability of FETCH is proven by the following pre-training experiments. The data-driven attention encoder module of the policy network in FETCH has the capability to encode the context of permutation-invariant and variable-length feature set of tabular data, which yields the probability of transferring prior knowledge across different datasets.

To verify the feasibility of transferability, we attempt to train FETCH on a range of datasets and save parameters of the attention encoder module as a pre-trained model. And apply this model instead of the initial random weights to the search of the new dataset, and see if it takes less time to achieve the same performance or better performance with the same number of exploration epochs. We define 4 kinds of pre-trained methods of FETCH. Each of them keeps the same initial parameters, except for the pre-trained datasets.

**No-Pre** means the model without any pre-training directly searching the FE plan from the scratch. **Pre-Oml** and **Pre-Uci** are the models pre-trained on 5 datasets from OpenML and UCI repository respectively (see Appendix B.4 for details). **Pre-Mix** is the model pre-trained on the whole 10 datasets of **Pre-Oml** and **Pre-Uci**.

We use these 4 pre-trained models to search FE plans on 5 previously unobserved datasets respectively, i.e. *Housing Boston*, *Openml_616*, *Openml_637*, *Ionosphere*, and *Wine Quality Red*. With the features transformation plan $\mathcal{T}$ of each dataset, the evaluation score is the performance of a learning algorithm $\mathbf{L}$ trained on the transformed feature set $\hat{\mathbf{X}}_T$ by 5-fold cross-validation.

Table 2 demonstrates the feasibility of transferability, where we compare the score of final top-1 FE plans explored by model searched from scratch (**No-Pre**) and the best model (**Pre-Best**) from the above 3 pre-trained models. And the maximum difference (Max-Diff) of improvement percentage between the pre-

Table 2: Transferability comparison of the best pre-trained model (Pre-Best) and non-pre-trained one (No-Pre). See text for details.

| Dataset | No-Pre | Pre-Best | Max-Diff (%) | Epoch$_{Max-Diff}$ |
| --- | --- | --- | --- | --- |
| Housing Boston | 0.5224 | **0.5357** | 2.31 | 123 |
| Openml_616 | 0.6749 | **0.6942** | 4.67 | 27 |
| Openml_637 | 0.6453 | **0.6631** | 5.51 | 26 |
| Ionosphere | 0.9716 | **0.9864** | 1.12 | 148 |
| Wine Quality Red | 0.6042 | **0.6207** | 2.59 | 152 |

trained models and the original model during the exploration and the number of epochs when they appear (Epoch$_{Max-Diff}$) are also listed in Table 2. It can be seen that the best FE plan found by

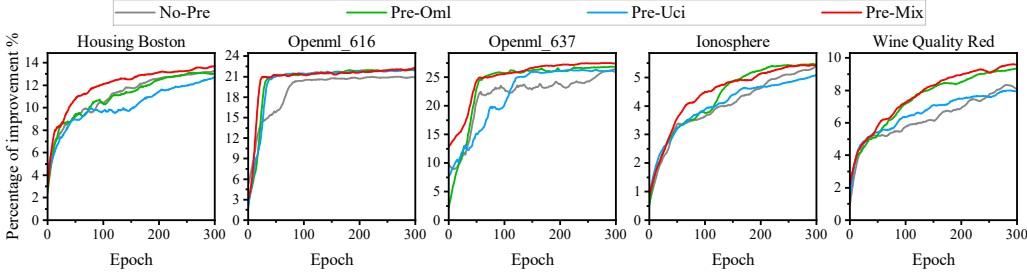

Figure 4: Transferability comparison of improvement (%) on 6 datasets under the model from scratch (No-Pre) and 3 kinds of pre-training models (Pre-Oml, Pre-Uci, and large-scale Pre-Mix). Better view in color.

the pre-trained approaches scored higher than the original model. Roughly the difference in the scores of the searched FE plans between the pre-trained model and the original one reaches its peak at around 150 epochs (half of the total epochs).

To further investigate the performance of the pre-trained model, Figure 4 visualize the smoothed curves of percentage of improvements in evaluation scores during searching under different types of pre-trained FETCH. In that, we find that the **Pre-Mix** model has higher scores than others, especially **No-Pre** (no pre-training), from the beginning to the end of the search in most cases. Other pre-trained models (i.e. **Pre-Oml** and **Pre-Uci**) share a similar trend, although their performance is not stable. The gap between **Pre-Mix** and **No-Pre** is larger in the early stage of the search, and **No-Pre** gradually chases up as the number of epochs increases. However, eventually the pre-training approaches still score higher than the original model.

These results suggest that the pre-training approach can be effective in improving scores and finding more appropriate feature engineering actions faster. Pre-training on data-driven FETCH can accumulate and transfer prior knowledge to unobserved datasets and improve FE efficacy more effectively. Despite experimental serendipity, the above experimental data still indicate to some extent the feasibility of simulating the knowledge transferability of human experts through pre-training AutoFE.

## 4.4 FLEXIBILITY TOWARD MODEL CHOICES

In this section, we measure the flexibility of using different ML models. We chose Logistic Regression (Wright, 1995), Random Forest (Liaw et al., 2002) and Xgboost (Chen & Guestrin, 2016) model for boosting comparisons on several datasets. We compare these models on the situation of no feature engineering (Base) and doing feature engineering with FETCH. As

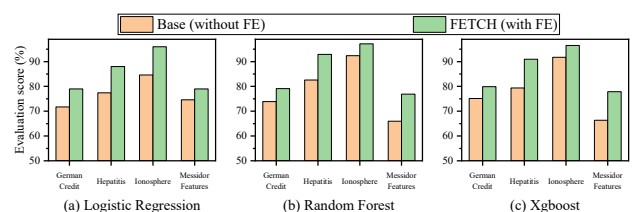

Figure 5: Flexibility comparison of evaluation scores (%) on different datasets and models.

shown in Figure 9, our framework has the effect of promoting the fitting effect for all these models. This also reveals the excellent flexibility of FETCH to different kinds of machine learning models.

## 5 CONCLUSION AND OUTLOOK

In this work, we propose FETCH, a novel RL-based end-to-end framework developed for AutoFE. The pivotal design of FETCH is to treat the tabular data as a "set" and use the datasets themselves as the *state* formulation in the MDP setup. The resultant framework achieves (surpassing or on-par) state-of-the-art performances on the standardized benchmarks adopted by prior work. Thanks to the methodological and architectural design of FETCH, we further concept-prove the feasibility of pre-training and transferability schemes for the tabular data. In summary, our work is to FETCH insights from data for better feature engineering and data mining. In the future, we hope to pre-train extremely large-scale models across various tabular data.

ACKNOWLEDGMENTS

This work is majorly supported by the National Key Research and Development Program of China (No. 2022YFB3304101), and in part by the NSFC Grants (No. 62206247) as well as the Key Research and Development Program of Zhejiang Province of China (No. 2021C01009). LL, HW, and JZ also thank the sponsorship by CAAI-Huawei Open Fund and the Fundamental Research Funds for the Central Universities.

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

# A  DETAILS OF RL SETUP

## A.1  MDP MODEL FOR FE

As discussed above, the feature engineering (FE) process conforms well to the exploration and exploitation paradigm, and is also a trial-and-error process. Therefore, the Markov decision process (MDP) can be employed to automate the simulation of the FE control problem. Also,the the goal of reinforcement learning (RL) is to find the optimal *policy* with the given MDP by an *agent*, where the policy is the mapping from *states* to *actions*, so that the final cumulative *reward* is maximized.

We unify the general problems of the FE process by casting it into an episodic MDP defined by a tuple $(\mathcal{X}, \mathcal{A}, \xi, \mathcal{R}, \mathbf{X}_0)$, where an agent $\mathcal{A}$ interacts with the environment $\xi$ which receives the agent's transformation actions and returns the transformed tabular data from initial data $\mathbf{X}_0$. Intuitively, in our case, the set of actions is the set of all possible (and supported) FE operations. Here, we define the variable $\mathcal{X} = \mathbf{X}_i$ as the *state* of MDP which is identical to the raw or transformed tabular data. In a single training step $i$, the agent receives the current state (feature set) $\mathbf{X}_i$, next it is required to choose a plan of feature transformation actions to synthesize the table data $\mathbf{X}_{i+1}$. Then the environment evaluates $\mathbf{X}_{i+1}$, and returns a reward $\mathcal{R}$ to update the agent and transmit it to a new state.

Next, we define the underlying concepts of MDP in the FE context, namely action space, state, agent, environment, and reward.

**FE Action Space**    Analogous to prior work (Zhu et al., 2022; Chen et al., 2019), our overall operation set (the space of action $t$) includes the following[6]:

- **The unary operation**: 6 lower-order value-conversion functions including `abs`, `square`, `inverse`, `log`, `sqrt`, `power3`.

- **The binary operation**: 4 combinatorial arithmetic functions including `addition`, `subtraction`, `multiplication`, and `division`.

Besides the operators on the numeric features, we also include operators for categorical ones:

- **Cross-combine operation** means to cross-product two categorical feature vectors to generate a combined one.

- **Binning operation** means discretizing numerical features by binning to further combine with categorical features.

As a framework that integrates feature generation and feature selection, FETCH also includes feature selection operators:

- **None** means taking no action on the specific feature.

- **Terminate** means stopping generating higher-order features based on the specific feature.

- **Delete** means deleting the specific feature from the original dataset.

The resulting FETCH framework is quite universal in terms of the types of datasets as well as their feature column properties. FETCH supports operations including unary, binary, selection operators, etc. Beyond, FETCH can easily integrate user-defined feature engineering operators that achieve a certain level of extensibility.

**State**    One core innovation of FETCH is our *data-driven* MDP setup where it parametrizes the tabular data both prior- and post-transformation into the *state* formulation. The details on how the agent "sees" the tabular data itself as a state are in Section 3.3. We argue that this further bridges the AutoFE with human experts who determines the next action move by scrutinizing the data rather than past actions sequence like prior work.

---

[6]For the sake of fairness, operators like "groupby" and "mean" are not added but are actually compatible with FETCH.

**The agent and policy network**    The agent is modeled by a policy network $\pi\colon \mathcal{X} \to P(\mathcal{A})$ where $P$ is the probability of the actions sampled by the agent $\mathcal{A}$ and $\mathcal{X}$ is the state parametrized by tabular data.

**The environment**    The environment $\xi$ encompasses the machine learning algorithm $\mathbf{L}$ with pre-configured hyperparameters and the evaluation metric $\mathbf{E}$. The interaction between the actions and the environment is intuitive; that is, FETCH performs feature engineering guided by the set of predicted actions on the processed feature columns, and runs it through $\mathbf{L}$ and $\mathbf{E}$. This induces the definition of the reward function.

**Reward Function**    It is crucial to get a precise reward estimation for training FETCH. In particular, our reward calculation process consists of an evaluation metric $\mathbf{E}$ related to a pre-selected model $\mathbf{L}$ (hyperparameters fixed), for example, a Logistic Regression classifier or a Random Forest predictor.

Similar to (Zhu et al., 2022; Chen et al., 2019; Khurana et al., 2018), we employ a standardized cross-validation strategy to enhance the precision of the reward estimation. The prior work generally adopts a most straightforward reward definition by using the average performance obtained on the k-fold split. However, we empirically observe that this causes great training instability, especially from the early stages. We conjecture that this is because the average operation has lost some non-neglectable information. For instance, a policy is optimal and needs to be reinforced if all k-fold exhibits superior performance. Nevertheless for the situations when the policy disagrees with the k-fold evaluation, we still need to punish the model according to the suboptimal results. Formally, the reward calculation can be written as:

$$\mathcal{R}(\mathbf{X}_i) = \bar{E}(\mathbf{X}_i) + E_{\text{diff}}(\mathbf{X}_i) \tag{2}$$

where $\bar{E}$ represents the average performance obtained from k-fold cross-validation and

$$E_{\text{diff}}(\mathbf{X}_i) = \sum_k \min(0, E_k(\mathbf{X}_i) - \bar{E}(\mathbf{X}_{i-1})) \tag{3}$$

where $E_k(\mathbf{X}_i)$ represents the k-th fold evaluation result, $\bar{E}(\mathbf{X}_{i-1})$ as the average performance from the tabular data from the previous step. Notably, this reward formulation prompts the agent to punish inferior results appearing in any fold, rather than simply looking at the average performance (Zhu et al., 2022). We found this means of reward calculation quite beneficial for training stability, especially in the early training epoch, and potentially leads to better sample efficiency.

## A.2    SAMPLING TRANSFORMATION ACTIONS

A final softmax is placed at the top stage after the policy network to obtain transformation actions probability, and further sample according to them to get actions plan $\mathcal{T} = \{t_1, t_2 \ldots t_m\}$:

$$\mathcal{T} \sim sample(\text{softmax}(\mathbf{h})) \tag{4}$$

where $\mathbf{h}$ denotes the hidden representation fed into the softmax layer. During the exploration, we obtain a stochastic action set in correspondence to the original feature sets by sampling actions through a multi-nomial distribution, provided by the above softmax operator. Then FETCH comprehensively evaluate the entire feature set generated by the actions set. This takes into account the overall performance of the entire feature set instead of the assumption in previous work that only evaluates a single performance improvement for each feature.

## A.3    TRAINING POLICY NETWORK WITH PPO

Proximal Policy Optimization (PPO) (Schulman et al., 2017) has been successfully applied to train RL agents in several fields. In FETCH, we employ the PPO algorithm to train the overall system. It allows our framework to sample several sets of action plans in each epoch, assign them to multiple threads (or workers) for parallel evaluation, and later compute the policy gradient to update the network. This way reduces sample complexity, which is also a kind of exploration cost, so that the policy network can be converged with fewer training epochs. The overall training algorithm of FETCH is shown as Algorithm 1 and the following.

---

**Algorithm 1** Training algorithm of FETCH

---

**Input:** Raw tabular data $\mathbf{X}_0$, policy network $\pi$, agent $\mathcal{A}$, metric $\mathbf{E}$, pre-selected ML model $\mathbf{L}$, label $\mathbf{Y}$

**Parameter:** Total exploration epochs $N$, maximum step (or order) to generate features $K$, number of workers (or threads) $W$

**Output:** New dataset $\hat{\mathbf{X}}_{\mathcal{T}}$ generated by the transformation $\mathcal{T}$ satisfying Equation 1

 1: **while** epoch $e \in [0, N]$ **do**
 2:    **while** worker index $w \in [0, W]$ **do**
 3:       Initiate worker[$w$]
 4:       **while** step $i \in [0, K]$ **do**
 5:          $\mathcal{X} = \mathbf{X}_i$
 6:          Sample a sequence of actions plan $\mathcal{T}_i$ through the
              $\pi \colon \mathcal{X} \rightarrow P(\mathcal{A})$
 7:          Obtain the generated tabular data $\mathbf{X}_{i+1}$ by letting the actions $\mathcal{T}_i$ interact with $\mathbf{X}_i$
 8:          Evaluate $\mathbf{X}_{i+1}$ by $\mathbf{E}(\mathbf{L}(\mathbf{X}_{i+1}, \mathbf{Y}))$
 9:          $i = i + 1$
10:       **end while**
11:       $w = w + 1$
12:    **end while**
13:    Calculate the reward $\mathcal{R}$ based on Equation 2
14:    Compute gradient based on the proximal policy optimization (Schulman et al., 2017) algorithm
15:    Update the policy network
16:    $e = e + 1$
17: **end while**

---

$r_\theta$ is defined as the ratio of sampling importance between new policy $\pi_\theta$ and old policy $\pi_{\theta'}$, it can be computed as follows.

$$r_\theta = \frac{\pi_\theta\left(a_t|s_t\right)}{\pi_{\theta'}\left(a_t|s_t\right)} \tag{5}$$

$C$ is defined as the advantage function $A^\theta\left(s_t, a_t\right)$ with the clip method.

$$\text{clip}\left(r_\theta, 1-\varepsilon, 1+\varepsilon\right) = \begin{cases} 1+\varepsilon, & r_\theta > 1+\varepsilon \\ r_\theta, & 1-\varepsilon \le r_\theta \le 1+\varepsilon \\ 1-\varepsilon, & r_\theta < 1-\varepsilon \end{cases} \tag{6}$$

$$C(s_t, a_t) = \text{clip}\left(r_\theta, 1-\varepsilon, 1+\varepsilon\right) A^\theta\left(s_t, a_t\right) \tag{7}$$

What needs a special explanation is that in our paper, we take the moving average of the past reward as the baseline for each time step. We compute the difference between the current reward and the baseline as the value of the reward function. The loss function is defined as the following formula.

$$J_{Policy}(\theta) \approx \sum_{(s_t, a_t)} \min\left(r_\theta A^\theta\left(s_t, a_t\right), C(s_t, a_t)\right) \tag{8}$$

To maintain exploratory properties, we add entropy $EN_{loss}$ to the loss function. The objective function of PPO can be summarized as follows:

$$J_{PPO}(\theta) = J_{Policy}(\theta) + EN_{loss}(\theta) \tag{9}$$

## B  DETAILS OF EXPERIMENTS

### B.1  ENVIRONMENTS

All experiments are carried out on a server with Ubuntu 20.04.1 LTS, Nvidia GeForce RTX 3090 (24GB GPU memory), Intel(R) Xeon(R) CPU (Gold 5218R CPU @ 2.10GHz, 64 cores), 256GB memory and 1TB hard drive. All experimental results are run with open-source code under the environment of Python 3.8.

## B.2 Hyperparameters

RL-agent learning rate $lr = 0.001$, discount factor $\gamma = 0.95$. The hyperparameters of Multi-Head Attention in the policy network are as follows, $d_{model} = 64$, $n_{head} = 6$, $d_v = 32$, $d_k = 32$. The maximum number of search epochs $N$ is limited to 300 (including DIFER and FETCH). Due to the requirements of NFS in their paper, we set $N$ to 100 epochs for it. The number of sampling also parallelized workers per round, is $W = 24$. The maximum feature order $K$ is set by $K = 3$. Other methods are limited to run for 5 hours respectively, which is the average running time of FETCH. All methods take their default parameters wherever possible.

We choose DIFER mostly because of its state-of-the-art performance on many benchmarks, while NFS is picked as a major rivalry due to the primary comparison between the MDPs. As being identical to their setup, we employ Random Forest (Liaw et al., 2002) as the pre-selected model in Experiment 4.2, 4.3 and Appendix C.6. All the hyperparameters of Random Forest (except $n\_estimator = 10$) are preset by default, and a stratified 5-fold cross-validation protocol is used to evaluate the effect of the selected feature engineering actions. In addition, we employ Logistic Regression (Wright, 1995), Xgboost (Chen & Guestrin, 2016), LightGBM (Ke et al., 2017) and Catboost (Prokhorenkova et al., 2018) as supplementary evaluation algorithms in Experiment 4.4 and Appendix C.5.

## B.3 Datasets

- Airfoil: https://archive.ics.uci.edu/ml/datasets/airfoil+self-noise

- BikeShare DC: https://www.kaggle.com/itssuru/bike-sharing-system-washington-dc

- Housing Boston: https://archive.ics.uci.edu/ml/machine-learning-databases/housing/

- House King County: https://www.kaggle.com/datasets/harlfoxem/housesalesprediction

- Openml_586: https://www.openml.org/d/586

- Openml_589: https://www.openml.org/d/589

- Openml_607: https://www.openml.org/d/607

- Openml_616: https://www.openml.org/d/616

- Openml_618: https://www.openml.org/d/618

- Openml_620: https://www.openml.org/d/620

- Openml_637: https://www.openml.org/d/637

- Adult Income: https://archive.ics.uci.edu/ml/datasets/adult

- Amazon Employee: https://www.kaggle.com/c/amazon-employee-access-challenge/data

- Credit Default: https://archive.ics.uci.edu/ml/datasets/default+of+credit+card+clients

- Credit_a: https://archive.ics.uci.edu/ml/datasets/Credit+Approval

- Fertility: https://archive.ics.uci.edu/ml/datasets/Fertility

- German Credit: https://archive.ics.uci.edu/ml/datasets/statlog+(german+credit+data)

- Hepatitis: https://archive.ics.uci.edu/ml/datasets/hepatitis

- Ionosphere: https://archive.ics.uci.edu/ml/datasets/ionosphere

- Lymphography: https://archive.ics.uci.edu/ml/datasets/Lymphography

- Megawatt1: https://www.openml.org/d/1442

- Messidor Features: https://archive.ics.uci.edu/ml/datasets/Diabetic+Retinopathy+Debrecen+Data+Set

- PimaIndian: https://www.kaggle.com/uciml/pima-indians-diabetes-database

- SpamBase: https://archive.ics.uci.edu/ml/datasets/Spambase

- SpectF: https://archive.ics.uci.edu/ml/datasets/SPECTF+Heart

- Wine Quality Red & White: https://archive.ics.uci.edu/ml/machine-learning-databases/wine-quality/

- Medical Charges: https://www.openml.org/d/42559

- Give Me Some Credit: https://www.kaggle.com/competitions/GiveMeSomeCredit/data

- Poker hand: https://archive.ics.uci.edu/ml/datasets/Poker+Hand

### B.4 PRE-TRAINED MODELS

- **No-Pre**: A model without any pre-training directly searching FE plan from the scratch.

- **Pre-Oml**: A model pre-trained on 5 datasets from OpenML. They are $Openml\_586$, $Openml\_589$, $Openml\_607$, $Openml\_618$, and $Openml\_620$.

- **Pre-Uci**: A model pre-trained on 5 datasets from the UCI repository, i.e. $Bikeshare\ DC$, $Credit\_a$, $SpamBase$, $Wine\ Quality\ White$, and $Credit\ Default$.

- **Pre-Mix**: A model pre-trained on the whole 10 datasets of **Pre-Oml** and **Pre-Uci**.

## C ABLATION STUDY

### C.1 EFFICIENCY OF FETCH

To measure the efficiency of FETCH, we plot the relationship between the improvements owing to feature engineering versus the exploration rounds, in Figure 6. As we can see the FETCH framework can generally obtain decent results by 100-150 epochs of exploration and then stably converges. Further, as a comparison with NFS, we count the feature evaluation times and use them as a proxy for a sample complexity comparison. The main rationale is that this process usually takes up a majority of the total running time in each round. Notably, FETCH requires around 15,000 times to converge while the NFS needs 160,000 times (Zhu et al., 2022). This indicates that FETCH is about 11x more sample efficient than its counterpart, which may further concept-proof our data-driven setup for MDP.

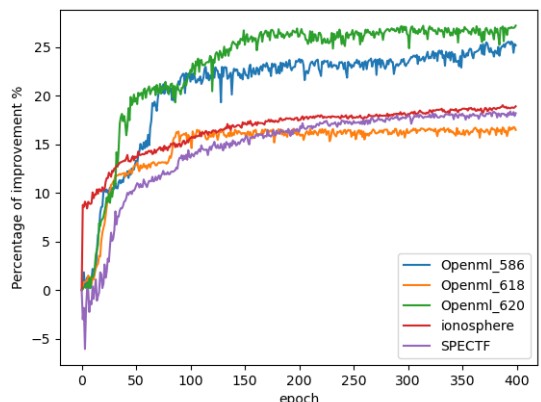

Figure 6: Exploration promotion process of several data sets, little improvement after 150 rounds, which means the policy network is converged.

### C.2 INFLUENCE OF HIGHER-ORDER FEATURES

In this section, we evaluate the influence of higher-order features extracted by FETCH on the eventual results. Figure 7(a) and Figure 7(b) show the effects of our algorithm on the Random Forest (RF) and Logistic Regression (LR) models, respectively. The vertical axis represents the percentage of improvement, and the horizontal axis represents the order. According to the experimental results, we get two conclusions: (i)-FETCH can significantly improve the score of both the two models, especially LR; (ii)-higher-order search leads to better performance for the LR model but is not guaranteed for RF.

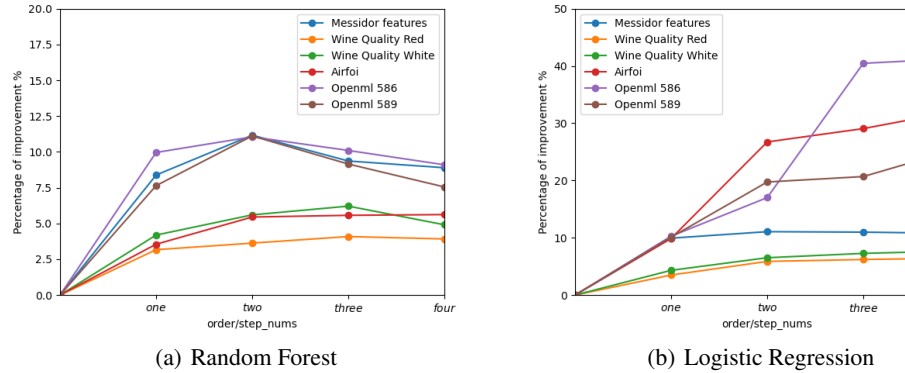

(a) Random Forest          (b) Logistic Regression

Figure 7: Effect of low/high order on different models. Apply the top-1 action series $\mathcal{T}$ to generate features and evaluate.

Figure 8 shows the proportion of high and low-order features. We stipulate that the second-order and above features are higher-order features. It can be seen from the figure that the proportion of higher-order features is very high.

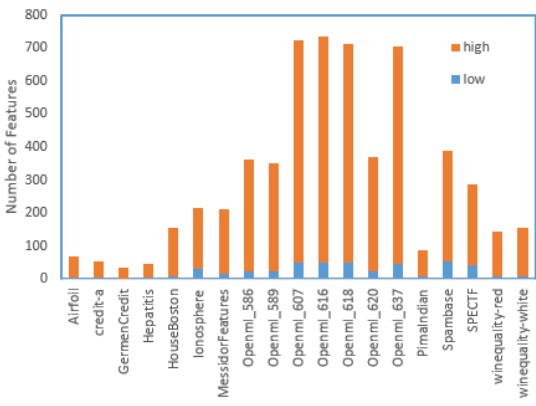

Figure 8: Low/high order features proportions of the experimental feature sets.

## C.3 SCALABILITY COMPARISON UNDER LIMITED TIME

Since the RL agent of FETCH only performs simple inference and data processing, empirically, the agent itself runs in a much smaller percentage of time than the training during downstream ML model evaluation. Thus,

Table 3: Scalability comparison of FETCH and other RL-based AutoFE methods on large-scale datasets under limited time (2 days).

| Dataset | C/R | Instances\Features | Base | NFS | DIFER | FETCH |
|---|---|---|---|---|---|---|
| Medical Charges | R | 163065\11 | 0.8821 | 0.8831 | 0.8832 | **0.8834** |
| Give Me Some Credit | C | 150000\10 | 0.9329 | 0.9339 | 0.9340 | **0.9341** |
| Poker hand | C | 1025010\10 | 0.6907 | 0.9259 | 0.9647 | **0.9971** |

as with the time-consuming model validation in NAS, the time performance of ML models is a bottleneck in increasing the speed of exploration in FETCH. We now evaluate the scalability of our approach on large-scale data (more than 50K rows) in Table 3. As a comparison, we use RL-based AutoFE approaches (i.e. NFS and DIFER) for our experiments. We compare the highest scores that can be achieved by different approaches to explore FE plans with the same *runtime (2 days)*, hardware resources and software environment. The final score is evaluated by 5-fold cross-validation after applying the highest-score FE actions plan on the entire original

dataset. The experimental datasets are from actual life scenarios. Table 3 demonstrates that our FE constructed on large-scale datasets can also lead to score improvements, which yields the capability for scalability at the same scale of running time.

## C.4 TIME EFFICIENCY COMPARISON

This section is an extended comparison of Experiment 4.2 in terms of time efficiency. Table 4 shows the total running time (in minutes) of several AutoFE methods on the above datasets. The results show that the time efficiency of the three AutoFE methods varies on different datasets. The overall time efficiency of each method is on the same scale. DIFER, as stated in their paper, outperforms the RL-based AutoFE in time efficiency. But FETCH can achieve higher scores.

Table 4: Time efficiency comparison of FETCH with other AutoFE methods. Here is the total execution time (in minutes).

| Dataset | C/R | Instances\Features | Execution Time | | |
|---|---|---|---|---|---|
| | | | NFS | DIFER | FETCH |
| Airfoil | R | 1503\5 | 50 | 85 | 70 |
| BikeShare DC | R | 10886\11 | 652 | 217 | 434 |
| Housing Boston | R | 506\13 | 54 | 154 | 158 |
| House King County | R | 21613\19 | 5098 | 602 | 939 |
| Openml_586 | R | 1000\25 | 329 | 167 | 192 |
| Openml_589 | R | 1000\25 | 326 | 176 | 197 |
| Openml_607 | R | 1000\50 | 1171 | 288 | 1383 |
| Openml_616 | R | 500\50 | 558 | 200 | 322 |
| Openml_618 | R | 1000\50 | 1224 | 238 | 365 |
| Openml_620 | R | 1000\25 | 318 | 70 | 181 |
| Openml_637 | R | 500\50 | 552 | 158 | 300 |
| Adult Income | C | 48842\14 | 897 | 141 | 300 |
| Amazon Employee | C | 32769\9 | 379 | 237 | 187 |
| Credit Default | C | 30000\25 | 1622 | 205 | 1246 |
| Credit_a | C | 690\6 | 59 | 116 | 99 |
| Fertility | C | 100\9 | 53 | 133 | 62 |
| German Credit | C | 1001\24 | 69 | 130 | 158 |
| Hepatitis | C | 155\12 | 50 | 90 | 96 |
| Ionosphere | C | 351\34 | 92 | 106 | 248 |
| Lymphography | C | 690\6 | 50 | 142 | 122 |
| Megawatt1 | C | 4900\12 | 92 | 85 | 256 |
| Messidor Features | C | 1150\19 | 73 | 116 | 160 |
| PimaIndian | C | 768\8 | 55 | 115 | 59 |
| SpamBase | C | 4601\57 | 439 | 117 | 800 |
| SpectF | C | 267\44 | 110 | 182 | 238 |
| Wine Quality Red | C | 999\12 | 61 | 292 | 76 |
| Wine Quality White | C | 4900\12 | 1622 | 205 | 114 |
| Average Rank | | | 2.0 | 1.7 | 2.3 |

## C.5 FLEXIBILITY TOWARD MORE MODEL CHOICES

This section is a supplement to the experiments in Section 4.4. We measure the flexibility of FETCH using two more ML models, i.e. LightGBM (Ke et al., 2017) and Catboost (Prokhorenkova et al., 2018), for boosting comparisons on several datasets. We compare these models on the situation of no feature engineering (Base) and doing feature engineering with FETCH. As shown in Figure 9, our framework has the effect of promoting the fitting effect for all these models. This also reveals the excellent flexibility of FETCH to different kinds of machine learning models.

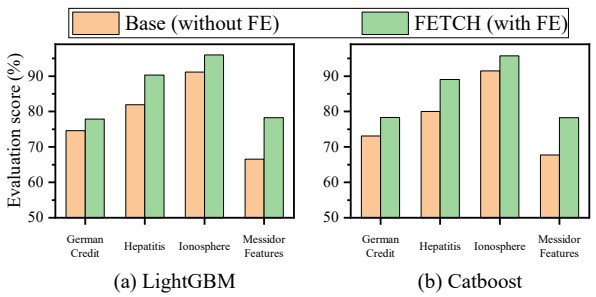

Figure 9: Flexibility comparison of evaluation scores (%) on different datasets and models.

## C.6 EFFECTIVENESS COMPARISON ON TEST SET

It is important to note that prior work such as NFS (Chen et al., 2019) and DIFER (Zhu et al., 2022) was coded with the goal of continuously improving the cross-validation scores over the entire dataset, i.e., not holding out the test set. We adopt this setup in Table 1 of Experiment 4.2, and

Table 5: Effectiveness comparison of FETCH with other AutoFE methods on test set. Underline means highest cross-validation score on training set. **Bold** indicates superior test score amongst AutoFE methods. "CV" denotes the highest cross-validation score obtained on the training set, and "Test" denotes the test score of the corresponding feature engineering plan on the test set.

| Dataset | Base CV | Base Test | DFS CV | DFS Test | AutoFeat CV | AutoFeat Test | NFS CV | NFS Test | DIFER CV | DIFER Test | FETCH CV | FETCH Test |
|---|---|---|---|---|---|---|---|---|---|---|---|---|
| Airfoil | 0.7119 | 0.7360 | 0.7338 | 0.7477 | 0.7307 | 0.7380 | 0.7424 | 0.7424 | 0.7367 | 0.7584 | 0.7572 | **0.7694** |
| Bikeshare DC | 0.9895 | 0.9901 | 0.9994 | 0.9993 | 0.9918 | 0.9922 | 0.9992 | 0.9993 | 0.9937 | 0.9945 | 0.9997 | **0.9997** |
| House King County | 0.6737 | 0.6842 | 0.6762 | 0.6884 | 0.6725 | 0.6812 | 0.6751 | 0.6879 | 0.6903 | **0.6943** | 0.6838 | 0.6844 |
| Housing Boston | 0.5992 | 0.6552 | 0.5938 | 0.6638 | 0.5982 | 0.6750 | 0.6353 | 0.6622 | 0.6310 | 0.6827 | 0.6576 | **0.6891** |
| Openml_586 | 0.6218 | 0.6594 | 0.6616 | 0.7189 | 0.6807 | 0.7153 | 0.7173 | 0.7310 | 0.7216 | 0.7369 | 0.7473 | **0.7688** |
| Openml_589 | 0.6298 | 0.6296 | 0.6587 | 0.6906 | 0.6598 | 0.6937 | 0.6804 | 0.6868 | 0.7033 | 0.6979 | 0.7189 | **0.7368** |
| Openml_607 | 0.6116 | 0.5923 | 0.6534 | 0.6172 | 0.6657 | 0.6201 | 0.6371 | 0.5780 | 0.6699 | 0.6365 | 0.7225 | **0.6860** |
| Openml_616 | 0.5073 | 0.5251 | 0.5019 | 0.5600 | 0.5620 | 0.5578 | 0.5390 | 0.5477 | 0.5637 | 0.5187 | 0.6275 | **0.6442** |
| Openml_618 | 0.5931 | 0.6470 | 0.6366 | 0.6397 | 0.6523 | 0.6860 | 0.6490 | 0.6450 | 0.6552 | 0.6832 | 0.7086 | **0.7187** |
| Openml_620 | 0.5911 | 0.6634 | 0.6073 | 0.6556 | 0.6456 | 0.6771 | 0.6401 | 0.6925 | 0.6760 | 0.6896 | 0.7197 | **0.7515** |
| Openml_637 | 0.5199 | 0.4319 | 0.5242 | 0.4894 | 0.5726 | 0.4563 | 0.5217 | 0.4523 | 0.5528 | 0.4570 | 0.6586 | **0.5756** |
| Adult Income | 0.8464 | 0.8529 | 0.8478 | 0.8496 | 0.8488 | 0.8382 | 0.8563 | 0.8565 | 0.8556 | **0.8580** | 0.8561 | 0.8553 |
| Amazon Employee | 0.9436 | 0.9458 | 0.9430 | 0.9450 | 0.9423 | 0.9440 | 0.9462 | **0.9475** | 0.9464 | 0.9447 | 0.9466 | 0.9460 |
| Credit Default | 0.8045 | 0.8022 | 0.8034 | 0.8026 | 0.8065 | 0.8053 | 0.8092 | 0.8054 | 0.8113 | **0.8069** | 0.8110 | 0.8027 |
| Credit_a | 0.8506 | 0.8942 | 0.8465 | 0.8702 | 0.8403 | 0.8702 | 0.8817 | 0.8558 | 0.8817 | 0.8606 | 0.8983 | **0.8846** |
| Fertility | 0.8429 | 0.8667 | 0.8286 | 0.8667 | 0.8571 | 0.8000 | 0.9286 | 0.8000 | 0.9429 | 0.8333 | 0.9143 | **0.9000** |
| German Credit | 0.7386 | 0.7233 | 0.7586 | **0.7900** | 0.7543 | 0.7833 | 0.7857 | 0.7500 | 0.7829 | 0.7333 | 0.7871 | 0.7367 |
| Hepatitis | 0.7974 | 0.8085 | 0.8160 | 0.8085 | 0.8797 | **0.8723** | 0.9173 | 0.8511 | 0.8896 | 0.8085 | 0.9537 | 0.8298 |
| Ionoshpere | 0.9265 | 0.9245 | 0.9347 | 0.9057 | 0.9429 | 0.8962 | 0.9592 | 0.9151 | 0.9551 | 0.9340 | 0.9796 | **0.9434** |
| Lymphography | 0.7962 | 0.8667 | 0.7862 | 0.8000 | 0.8348 | 0.8222 | 0.8548 | 0.8444 | 0.8543 | 0.8444 | 0.9029 | **0.8667** |
| Megawatt1 | 0.8925 | 0.8816 | 0.8870 | 0.8816 | 0.8981 | 0.8947 | 0.9265 | 0.8684 | 0.9322 | 0.8684 | 0.9379 | **0.9211** |
| Messidor Features | 0.6534 | 0.6387 | 0.7006 | 0.7486 | 0.7106 | 0.6965 | 0.7416 | 0.7312 | 0.7528 | 0.7254 | 0.7677 | **0.7775** |
| PimaIndian | 0.7449 | 0.7056 | 0.7691 | 0.6883 | 0.7524 | 0.7229 | 0.7953 | 0.7403 | 0.7971 | **0.7446** | 0.8120 | 0.7359 |
| SpamBase | 0.9410 | 0.9464 | 0.9366 | 0.9435 | 0.9385 | 0.9522 | 0.9484 | 0.9522 | 0.9484 | 0.9522 | 0.9516 | **0.9522** |
| SpectF | 0.8276 | 0.7901 | 0.7905 | 0.8148 | 0.7743 | 0.8148 | 0.8602 | 0.8148 | 0.8708 | 0.8272 | 0.8818 | **0.8395** |
| Wine Quality Red | 0.6265 | 0.6563 | 0.6533 | 0.6625 | 0.6140 | 0.6667 | 0.6622 | 0.6458 | 0.6613 | 0.6625 | 0.6774 | **0.6667** |
| Wine Quality White | 0.6144 | 0.6347 | 0.6161 | 0.6429 | 0.6144 | 0.6347 | 0.6377 | 0.6299 | 0.6365 | 0.6442 | 0.6415 | **0.6463** |

the experimental results show the superiority of FETCH. However, this setup may be contrary to real-life application scenarios where we often need to explore good features on the training set and validate them on the test set, which is often used only once as the final evaluation metric, rather than repeatedly using the test set scores to optimize feature engineering.

To simulate the authentic situation, we split the original dataset into a training set and a test set in the ratio of 0.7/0.3 and let all AutoFE methods optimize the feature engineering plan on the training set to obtain higher cross-validation scores and apply it to the test set to obtain the final test scores. Unlike previous experiments using the default order of the datasets, in this experiment we shuffled the original data using the same random seeds, which may result in differences in scores of Table 1. The experimental results are displayed in Table 5.

The experimental results show that FETCH achieves both the highest cross-validation scores and test scores on 19/27 datasets. Empirically, an effective and robust feature engineering plan tends to achieve high scores on both the training and test sets. Therefore, the previous setup of directly optimizing the cross-validation scores is reasonable and can largely improve the generalization ability of the searched feature engineering plans.

# D DISCUSSION

## D.1 FEASIBILITY OF APPLYING FETCH TO OTHER FORMS OF DATA

We may generalize our framework to other domains by regarding the feature outputs of an encoder (e.g. convolutional neural networks for images, recurrent neural network for speech and graph neural network for graphs (Goodfellow et al., 2016)) as tabular inputs. However, vision and language tasks have a large amount of data collection sharing common sequential/spatial structures. In view of the rich literature on these data, we suppose existing full-fledged pre-trained feature extractors (ResNet (He et al., 2016), SimCLR (Chen et al., 2020), and BERT (Devlin et al., 2018)) would be a better choice. In contrast, tabular data has the properties of permutation invariance and variable length, and our FETCH is more suitable to tackle these.

Nevertheless, our approach also opens the door to learn from tabular-included multimodal data. For example, when the dataset contains both image and tabular attributes, we can concatenate pre-extracted image features with tabular inputs and run our FETCH model to obtain a better combination of features.

