# OpenReview forum: "Learning a Data-Driven Policy Network for Pre-Training Automated Feature Engineering"
_ICLR.cc/2023/Conference — ICLR 2023 notable top 25%_

### Official Review · Reviewer_B9Km · 2022-10-21

**Confidence:** 3
**Correctness:** 4
**Technical Novelty And Significance:** 4
**Empirical Novelty And Significance:** 4
**Recommendation:** 8

**Clarity, Quality, Novelty And Reproducibility:**

This paper is overall well-written and clear. The code in the supplementary material seems to be well-organized and I believe it works as the paper claims, although I have not actually verified it.



**Strength And Weaknesses:**

Strengths:
1. The proposed framework is novel, simple and effective, which is easily to follow. The pre-training technique seems like a new schema in the tabular data mining field.
2. The paper is well-written, and the related work is covered well.
3. The experiments are quite comprehensive. The authors also show that pre-training FETCH on more datasets results in additional performance gains, which further strengthens the claim of the model's transferability.

Weaknesses:
1. It is nice that the authors include 27 datasets as a benchmark and demonstrate that FETCH is much better than the other AutoFE baselines in most cases. However, I don't quite sure why the authors are comparing it to popular open-source AutoML packages (i.e. AutoSklearn and AutoGluon)? After all, these are two different domains. Is it to prove that AutoFE can power AutoML? Maybe I'm missing it, and please clarify.
2. Section 4.4 only covers 3 comparison algorithms, it would be better to add the enhancement effect on more algorithms such as LightGBM and CatBoost, which will be more convincing for the adaptability of FETCH on various algorithms.
3. FETCH seems to have been pre-trained on only 5 datasets, which seems somewhat simple. It would be preferable to have pre-training results on a larger number of datasets.
4. The theoretical analysis is a bit lacking. It is hoped that a theoretical analysis of effectiveness and experimental results can be provided as space permits.


**Summary Of The Paper:**

This paper firstly specifies the fact that AutoFE is underappreciated in the AutoML pipeline tailored to tabular data nowadays, and summarizes that previous AutoFE work faced two major drawbacks of unobserved data and lack of transferability. To tackle these problems, this paper proposes a novel RL-based AutoFE framework(called FETCH) by observing the behavior patterns of human experts, which is equipped with a data-driven MDP setup in its policy network. The key idea is to consider each table as a set of features and incorporate the table into the MDP state. Then FETCH learns how to map from data to feature engineering actions. Experiments show that FETCH outperforms other SOTA or popular feature engineering baselines and supports transferability by pre-training.

**Summary Of The Review:**

The idea of learning a data-driven policy network is well-motivated and novel, and the experiments are compelling.

---

> ### Author Response · Authors · 2022-11-15
> **Official Response to Reviewer B9Km**
>
> Thank you for the good words! As a briefing, we added the following section in our revision to respond to the reviews: (i) we have added comparative experiments for models LightGBM and CatBoost in Appendix C.4. We provide a more detailed explanation and responses as the following.
>
> **Q1:**
> > I don't quite sure why the authors are comparing it to popular open-source AutoML packages (i.e. AutoSklearn and AutoGluon)? After all, these are two different domains. Is it to prove that AutoFE can power AutoML?
>
> **A:**
> As discussed in Section 4.2, these two AutoML baselines focus primarily on model architecture/hyperparameters search and involve no feature engineering. We believe that these comprehensive comparisons will further highlight the effectiveness of FETCH and the boosting performance it can deliver, even beyond the scope of feature engineering.
>
> **Q2:**
> > Section 4.4 only covers 3 comparison algorithms, it would be better to add the enhancement effect on more algorithms such as LightGBM and CatBoost, which will be more convincing for the adaptability of FETCH on various algorithms.
>
> **A:**
> Thanks for your advice. We have added the comparison on LightGBM and CatBoost in Appendix C.4. FETCH still shows decent performance on both added algorithms ;)
>
> **Q3:**
> > FETCH seems to have been pre-trained on only 5 datasets, which seems somewhat simple. It would be preferable to have pre-training results on a larger number of datasets.
>
> **A:**
> Yes, indeed, five datasets for a pre-training seems overly simple. The main reason behind it is that we simply want to highlight the ability to transfer across tables from FETCH. Noted, this is unprecedented for tabular data feature engineering and modeling; FETCH can achieve it majorly thanks to its novel data-driven framework re-work.
> True that the pre-training normally requires a massive amount of datasets (such as BERT, GPT, MoCo and others). We simply want to concept-prove that FETCH can be utilized as a preliminary attempt for this line of pre-training research on tabular data. Hence a full large-scale pre-training may seem beyond the scope of this paper.
> Regardless, we hope to dig further into this line in the future. Of course, extending the five datasets to many is the very first step to take ;) Thank you!
>
> **Q4:**
> > The theoretical analysis is a bit lacking. It is hoped that a theoretical analysis of effectiveness and experimental results can be provided as space permits.
>
> **A:**
> Yes, the theoretical side of analysis or innovation is lacked from our paper.
> In our defense, the theories around feature engineering for tabular data is still a blank, by scrutinizing the whole spectrum of this domain since the very early years. However, we do agree with the reviewer that this aspect is important. We hope to study it in the near future.

---

### Official Review · Reviewer_Y4Ri · 2022-10-23

**Confidence:** 5
**Correctness:** 4
**Technical Novelty And Significance:** 3
**Empirical Novelty And Significance:** 4
**Recommendation:** 8

**Clarity, Quality, Novelty And Reproducibility:**

The paper is in general well written. There are a few loose points but nothing major.
It is easy to read and comprehend.
Reproducibility is questionable. There are not many details about how to reproduce the results.

**Strength And Weaknesses:**

Strengths:
The idea of using data directly is novel.
The use of data enables transfer learning.

Weaknesses:
Scalability is questionable. There is no deep discussion on computational times and comparisons with other algorithms.
Explainability/interpretability is questionable. To the contrary, the resulting training data is not interpretable.

**Summary Of The Paper:**

Automated feature engineering is another needed ingredient in 'data science as a service.' Prior work focuses either on feature selection or feature engineering that is independent on the underlying data. The authors propose an RL algorithm that directly transforms data by applying a sequence of unary or binary operations.


**Summary Of The Review:**

The underlying idea of using the actual data in the RL setting is very novel. It is a substantial contribution. In my opinion it outweighs deficiencies in the work.
Once the RL is established (the state space), everything else is standard (the network, algorithm).

The authors don't mention feature selection (forward/backward selection, etc) which is perhaps injustice. Maybe feature selection should also be benchmarks (perhaps autoML methods include them).

Minor remarks:
Page 1: next to the last paragraph states that there is significant prior autoFE work. This statement contradicts some prior statements stating there is not a lot of work in this area.
Hyperparameters being fixed is questionable since different subset of features definitely dictate possible different hyperparameters. I understand that this inclusion would be a significant complication and thus I don't take this against the authors.
Page 3: 'specified feature x' excludes binary operations. This statement should be reworded.
Page 4: weightheight^2 should be weightheight$^2$

---

> ### Author Response · Authors · 2022-11-15
> **Official Response to Reviewer Y4Ri**
>
> Thank you very much for your good words and suggestions! We are thrilled that you enjoyed our paper. We have updated our paper based on the suggestions and comments of the reviewers. Summary of revision: (i) we added scalability comparison in Appendix C.3. (ii) we added time efficiency comparison in Appendix C.4. (iii) we updated our supplementary material for more descriptive details.
>
> **Q1:**
> > Scalability is questionable. There is no deep discussion on computational times and comparisons with other algorithms.
>
> **A:**
> We have added a comparison of the effectiveness and computational time among Base, NFS, DIFER and FETCH on large-scale data (>50k instances) in Appendix C.3. Besides, we reported the running time of NFS, DIFER and FETCH in Appendix C.4. As can be concluded from these results, FETCH possesses the time complexity at same-scale with other approaches, but it relishes the significant performance boost.
>
> **Q2:**
> > Explainability/interpretability is questionable. To the contrary, the resulting training data is not interpretable.
>
> **A:**
> We wholeheartedly agree with the reviewer that the explainability/interpretability is quite important for tabular data modeling. As we pointed out in the paper, the goal of AutoFE is to explicitly generate new features using multiple features for computation. The new features will be named by the computational path. For example, when a Body Mass Index feature is generated, the resulting training data will contain a column named weight/height$^2$. This explicit way of showing new feature generation paths would be more interpretable than deep tabular models (like TabNet, AutoInt) due to their internal implicit feature intersections.
> In that regard, our tentative solution to the explainability/interpretability of FETCH is to: (i)-first delve into the downstream model such as XGBoost or a Random Forest and attain the ``feature_importance_`` score; (ii)-then based on our explicit feature generation tree, we could further decompose the feature importance downwards to the original feature of the given table input.
>
> **Q3:**
> > Reproducibility is questionable. There are not many details about how to reproduce the results.
>
> **A:**
> Sorry, our bad! As we stated in the opening general comment, we have added more experimental reproduction settings and updated the code. We may refer the reviewer for the Appendix B and the supplementary material for more descriptive details.
>
> **Q4:**
> > The authors don't mention feature selection (forward/backward selection, etc) which is perhaps injustice. Maybe feature selection should also be benchmarks (perhaps autoML methods include them).
>
> **A:**
> This is perhaps a writing confusion, our apologies!
>
> In effect, the AutoFE family mostly integrates the feature selection operations into the feature manipulation procedure, e.g., None, Terminate, Delete (see Appendix A.1). In other words, the feature selection has been encapsulated into the feature engineering process; namely in FETCH and all other competing methods.
>
> For the other point the reviewer raised, the AutoML method incorporated and mentioned in our paper does not implement feature generation or selection. Rather, these operations are ubiquitously pushed into the AutoFE procedure.
>
> **Q5:**
> > Minor remarks…
>
> **A:**
> Thanks a million for these detailed remarks. We have revised these problems in the new version.

---

### Official Review · Reviewer_vD6x · 2022-10-25

**Confidence:** 4
**Correctness:** 3
**Technical Novelty And Significance:** 3
**Empirical Novelty And Significance:** 3
**Recommendation:** 8

**Clarity, Quality, Novelty And Reproducibility:**

Novelty:
1. Paper proposes a new form of markov decision process to generate features. Previous methods take number of features and only update using sequences of previous actions, iteratively. Whereas Fetch provides Feature engineering actions and constructions actions, instead of features, iteratively based on the newly generated datasets (features).
2. According to this paper, Previous methods for AutoFE have never explored transferability between datasets, which has been done by this method.


Clarity & Quality: The paper has high clarity with explicit background provided and methods explained with sufficient depth both in terms of mathematical equations, datasets and analysis of results. There is further in-depth explanation in the appendix section around usage of material and datasets which is crystal clear.


Reproducibility: The method has been extensively tested on publicly available datasets which are linked. Algorithm and methods have been provided as well. The paper states that all experiments have been run with open source code provided.


**Details Of Ethics Concerns:**

None.

**Strength And Weaknesses:**

Strengths:
1. One of the key strengths of this paper is its ability to provide transferability between datasets which previous methods have not been able to provide.
2. The method is highly flexible as it has been applied on different datasets as well different ML models.
3. Overall there is sound discussion relevant work, gap analysis to find opportunity of development, strong mathematical rigour and experimental setup, evaluation and analysis.


Weaknesses:
1. The authors declare that to the best of their knowledge there aren't any autoFE/autoML workarounds for managing tabular data to accomplish transferability. However this could be supported by a stronger statement that can more concretely say if such methods exist or not by clearly stating a comprehensive survey of analysis did not yield any methods.
2. This method is specifically geared towards tabular datasets. While tabular datasets prevail in several key applications, it would be interesting to know how this method applies/does not apply to any other form of datasets (eg. images, speech, unstructured datasets). At least there could be some discussion on consideration, if not a full scale evaluation.


**Summary Of The Paper:**

This paper proposes a first of its kind architecture framework for automated feature engineering called Fetch, a system based on brand new data driven Markov decision process. The authors identify critical gaps by stating the current methods for AutoFe are insufficient when it comes resembling human effort in handling datasets as the underlying Markov decision setup is different i.e. more based on trial and error, which leads to poor generalization across datasets and higher computational costs. The new method also has a key element of transferability, which is basically the ability to enable feature engineering on new datasets using prior policy networks trained on previous datasets, without the need for additional data exploration. The authors present evidence that Fetch is superior to the existing state of the art automated feature engineering methods such as Random, DFS, AutoFeat, NFS, DIFER as well as AutoML methods such as AutoSklearn and AutoGluon. The method is also tested for transferability by application on several datasets.

The authors argue that the approach comes very close to mimicking human experts when it comes to handling new datasets when it comes to transferring experience.

**Summary Of The Review:**

I propose that we accept this paper on the basis of its merits around developing a new AutoFE method based on a novel markov decision process which not only is superior to the existing state of the art in terms of performance but also has a unique feature of transferability. Both strong points are well argued and evidenced in the paper throughout with sufficient gap analysis around opportunity areas, discussion of background and related work, strong mathematical rigor and sound experimental design setup, with thorough data analysis, discussion of evaluation and results. Although the work is limited to tabular data there is thoughtful coverage on several popular datasets including regression and classification examples and comparison to other AutoFE models and coverage across several ML models.

---

> ### Author Response · Authors · 2022-11-15
> **Official Response to Reviewer vD6x**
>
> Thanks very much for your insightful comments and suggestions! We have updated our paper based on the comments of the reviewers. Summary of revision: (i) we modified our statement about the existence of some transferability work; (ii) we have added a discussion of applying FETCH to other forms of datasets in Appendix D.1.
>
> **Q1:**
> > The authors declare that to the best of their knowledge there aren't any autoFE/autoML workarounds for managing tabular data to accomplish transferability. However this could be supported by a stronger statement that can more concretely say if such methods exist or not by clearly stating a comprehensive survey of analysis did not yield any methods.
>
> **A:**
> Thanks for the suggestion! Based on our extensive searching on the literature, while the transferability for modeling tabular data was briefly discussed by few prior work, we are certain that such transferability in the regime of feature engineering, AutoFE and AutoML is very much untapped. Parallel evidence can also be seen by [1].
> Regardless, we changed our wording as per your suggestion. Thanks again!
>
> **Q2:**
> > While tabular datasets prevail in several key applications, it would be interesting to know how this method applies/does not apply to any other form of datasets (eg. images, speech, unstructured datasets). At least there could be some discussion on consideration, if not a full scale evaluation.
>
> **A:**
> Our answer is yes!
> For example, we may generalize our framework to other domains by regarding the feature outputs of an encoder (e.g. CNN for images, RNN for speech and GNN for graphs) as tables. In contrast to the tabular data though, the vision and text data forms generally possess richer spatial or sequential structures. How to properly represent these structures and encode them into FETCH would be an interesting line of future work. FETCH, on the other hand, is devised to tackle the very characteristic of the tabular data including the permutation invariance or the variant length feature (see Section 3.3). In that regard, we have also added some discussion to Appendix D.1 of our revised manuscript.
>
> [1] Roman Levin, Valeriia Cherepanova, Avi Schwarzschild, Arpit Bansal, C Bayan Bruss, Tom Goldstein, Andrew Gordon Wilson, and Micah Goldblum. Transfer learning with deep tabular models. arXiv preprint arXiv:2206.15306, 2022.

---

### Public Comment · ~Tianping_Zhang1 · 2022-11-09
**Unable to reproduce the experiments**

Thanks for your interesting work. We are attracted by your work and try to run your models on our datasets. However, critical files are missing and the codes in your supplementary materials cannot run. For example, the ‘instance_selection’ file imported in main_attention.py is missing. In addition, it would be much appreciated if you could provide all the dependencies needed to run the codes as well as all the scripts and hyperparameters needed to reproduce the experimental results.

---

> ### Author Response · Authors · 2022-11-15
> **Response: Update of Supplementary Material**
>
> Our bad! This is caused by redundant lines in our code. With a deletion of those lines (Line 10, 88 and 89 in ``main_attention.py``), our code is ready-to-go. The code has been updated. We also add a new dependency file ``requirement.txt`` in the supplementary material.
>
> Hope this helps! Any further question, let us know.

---

### Public Comment · ~Mark_Wang1 · 2022-11-14
**From a concurrent submission to ICLR2023**

Thanks for the interesting work. We are coming from a concurrent submission to ICLR2023 on the topic of automated feature generation (https://openreview.net/forum?id=CnG8rd1hHeT&noteId=OhEP681JJL) (we used very different approach though). Our algorithm is very lightweight and efficient, so we quickly test our methods OpenFE on the datasets you provided in the supplementary materials. Surprisingly, we find out that OpenFE outperforms FETCH on 17 out of 20 datasets used in your paper. We have investigated the details of your supplementary materials and tried our best to ensure fair comparisons on the same experimental settings. We have released the codes in our supplementary materials, and you are welcome to reproduce our results (please refer to more details in our response to all reviewers https://openreview.net/forum?id=CnG8rd1hHeT&noteId=KihCSC53QS0). In addition, we cannot reproduce your results due to the absence of crucial files in the supplementary materials.

To abide by the ICLR anonymity rule, we asked a third party to post the comment for us (so Mark Wang is not a co-author).

---

> ### Author Response · Authors · 2022-11-15
> **Official Response to the cocurrent submission of OpenFE (1/3)**
>
> We thank Mark for serving as a friendly third-party bringing out attention to this concurrent work. In that, we form this following response of three-folds:
>
> (1)	Unfortunately, after multiple checks of our code and your provided code, we conclude that this experimental comparison that Mark posted is **not fair**.
>
> (2)	After we carefully set all the hyper- and meta-parameters in our code consistent with the provided code, our experimental conclusion is the opposite from what you have posted and **substantially better than OpenFE** under fair comparisons on 17 out of 20 datasets. The results listed below and our code updated on Openreview.
>
> (3)	We do have several **ethical concerns** towards this post, including anonymity leakage and misleading of the reviewing process.
>
> Here we post our detailed response.
>
> 1. About the fairness of comparisons: we found multiple setup differences that differing from FETCH --- as well as prior work like NFS, DIFFER and others --- that unfortunately leads to an unfair comparison.
>
> (a) Your paper used 40 operators as opposed to 11 operators in ours. While more operators are promising towards better modeling performance, we stuck FETCH with 11 operators because this is consistent with all prior work. Noted, we believe that the point of pushing AutoFE is more about its methodological innovation for ICLR, rather than engineering efforts like extending the operator library.
>
> (b) To further form fair comparisons, in your code, the downstream model adopts a random forest which is consistent with the rivaling methods. However, in this pack of code it doesn't have the proper hyper-parameter of ``n_estimator=10``. We understand this could be hyper-parameter tunning strategy. However, the AutoFE is much emphasized in the space of feature engineering, rather than the downstream modeling side. We kept FETCH **consistent everywhere with prior work**.
>
> (c) **A BIZZARE BUT CRUCIAL ONE**: the shuffling is turned True in your cross-validation setup. Yes, true that using a randomly shuffled cross-validation is quite common in ML applications. **However, in many of the dataset setups in this regime, we did find that shuffling the data in the stage of cross-validation could significantly bump the scores.** Therefore, for fair comparison with all prior work, we kept the dataset as is and unshuffled. The prior work ubiquitously stuck with this unshuffled version, identical to FETCH's setup.
>
> 2. With all that being said in 1, we re-ran your code with more rigorously checked consistency with FETCH, as well as all prior work for fair comparison. Namely, we made a few changes upon your provided code and we also uploaded this version of yours in this [Mega Cloud](https://mega.nz/file/Nbd2SD6L#1IfJKYgXnFrVC_vwrnHP-pY9KDNHylsbojSTZFQrhR8) (including `results.csv` counted by your code). Feel free to vimdiff them to check further. The changes include (i)-we change your action operator set to be back to 11 (in line 5 ~ 16 of  `/rebuttal/FeatureGenerator.py `); (ii)-we keep the random forest hyper-paramter consistent by setting ``n_estimator=10`` (line 12, 17 in `/rebuttal/feature_generation_and_selection.py `); (iii)-we turn off your cross-validation shuffling as identical with ours and all others (line 39, 43 in `/rebuttal/main.py`). The results are attached down below.
>
> We conclude from this table that **the conclusion from your previous post does not stand properly**. The comparative results still makes FETCH to stand out and the prior work is now compared fairly.
>
> Indeed, there are two holes to fill.
>
> On one thing, yes, **the shuffling aspect is disturbing**. We argue that this is not majorly the concern from developing AutoFE methodologies, or even AutoML methodologies. This is mostly about the nature of dataset stats, listing logics and others. Perhaps some of these datasets and listed following certain fixed logic and heuristic that fixing the order could be unfriendly for cross-validation modeling. However, **we do believe that keeping this setting consistent across all prior work is extremely substantial, especially on forming an appropriate and fair comparative testbed**.
>
> On the other hand, what we provided was to bring your setting closer to our setting (as well as all prior work). **What we are currently doing is to bring ours to yours**. Like we said, while this may put all prior work our of the cage due to unfair comparing issue, with all due respect we hope to bring new insights from this comparison. Since there are some coding work to do, we hope to get this result out soonish.

---

> ### Author Response · Authors · 2022-11-15
> **Official Response to the cocurrent submission of OpenFE (2/3)**
>
> The comparison results are attached down below.
>
> | Dataset            | #samples | #features | Base   | DFS    | AutoFeat | NFS    | DIFER  | FETCH       | Your Reported OpenFE | Our Reproduction of OpenFE | New Difference |
> | ------------------ | -------- | --------- | ------ | ------ | -------- | ------ | ------ | ----------- | -------------------- | -------------------------- | -------------- |
> | Airfoil            | 1503     | 5         | 0.5068 | 0.6003 | 0.5955   | 0.6226 | 0.6125 | **0.6463** | 0.7894               | 0.6089                     | 0.0374         |
> | BikeShare DC       | 10886    | 11        | 0.9880 | 0.9990 | 0.9891   | 0.9991 | 0.9995 | **0.9997** | 0.9998               | 0.9996                     | 0.0001         |
> | Housing Boston     | 506      | 13        | 0.4641 | 0.4708 | 0.4703   | 0.4977 | 0.5072 | **0.5224** | 0.6867               | 0.4789                     | 0.0435         |
> | House King County  | 21613    | 19        | 0.6843 | 0.6908 | 0.6917   | 0.6934 | 0.6948 | **0.7475** | 0.7359               | 0.7013                     | 0.0462         |
> | Openml_586         | 1000     | 25        | 0.6564 | 0.7188 | 0.7178   | 0.7223 | 0.6946 | **0.7671** | 0.7746               | 0.7274                     | 0.0397         |
> | Openml_589         | 1000     | 25        | 0.6395 | 0.6959 | 0.7278   | 0.7165 | 0.6789 | **0.7562** | 0.7488               | 0.7216                     | 0.0346         |
> | Openml_607         | 1000     | 50        | 0.6363 | 0.6815 | 0.6499   | 0.6485 | 0.6564 | **0.7404** | 0.7518               | 0.6987                     | 0.0417         |
> | Openml_616         | 500      | 50        | 0.5605 | 0.5807 | 0.5927   | 0.5856 | 0.5982 | **0.6749** | 0.6894               | 0.6099                     | 0.0650         |
> | Openml_618         | 1000     | 50        | 0.6351 | 0.6848 | 0.6374   | 0.6461 | 0.6553 | **0.7351** | 0.7472               | 0.6749                     | 0.0602         |
> | Openml_620         | 1000     | 25        | 0.6309 | 0.6528 | 0.6574   | 0.6943 | 0.7262 | **0.7506** | 0.7536               | 0.7063                     | 0.0443         |
> | Openml_637         | 500      | 50        | 0.5160 | 0.5105 | 0.5763   | 0.5739 | 0.6006 | 0.6453      | 0.6772               | **0.6530**                | -0.0077        |
> | Amazon Employee    | 32769    | 9         | 0.9492 | 0.9447 | 0.9499   | 0.9510 | 0.9504 | **0.9516** | 0.9737               | 0.9441                     | 0.0075         |
> | Credit_a           | 690      | 6         | 0.8044 | 0.8056 | 0.8086   | 0.8101 | 0.8108 | 0.8114      | 0.8829               | **0.8493**                | -0.0379        |
> | Fertility          | 100      | 9         | 0.8700 | 0.7900 | 0.8910   | 0.9189 | 0.8800 | **0.8900** | 0.9288               | 0.8100                     | 0.0800         |
> | Hepatitis          | 155      | 12        | 0.8258 | 0.8516 | 0.8677   | 0.8766 | 0.8839 | **0.9290** | 0.8862               | 0.7935                     | 0.1355         |
> | Messidor Features  | 1150     | 19        | 0.6594 | 0.7089 | 0.7359   | 0.7417 | 0.7541 | **0.7689** | 0.7693               | 0.7272                     | 0.0417         |
> | SpamBase           | 4601     | 57        | 0.9154 | 0.9198 | 0.9237   | 0.9341 | 0.9372 | 0.9405      | 0.9434               | **0.9450**                | -0.0045        |
> | SpecfF             | 267      | 44        | 0.7751 | 0.8125 | 0.8331   | 0.8608 | 0.8538 | **0.8838** | 0.8911               | 0.7901                     | 0.0937         |
> | Wine Quality Red   | 999      | 12        | 0.5597 | 0.5422 | 0.5641   | 0.5814 | 0.5779 | **0.6042** | 0.7023               | 0.5441                     | 0.0601         |
> | Wine Quality White | 4900     | 12        | 0.4976 | 0.4855 | 0.5023   | 0.5111 | 0.5153 | **0.5235** | 0.6866               | 0.4920                     | 0.0315         |

---

> ### Author Response · Authors · 2022-11-15
> **Official Response to the cocurrent submission of OpenFE (3/3)**
>
> 3. Unfortunately, we feel obliged to write our ethical concern down. Our concern is three-fold.
>
> (a)	To begin with, Openreview --- as a great paper submission platform --- is to get public cohorts to involved in the reviewing process and let them comment. It should NOT be used as to bias the judging process on two concurrent work submitted to the same venue. We should simply let the ACs and reviewers of each paper to judge on them respectively. We also saw the comment you made under your paper channel. We think it is perhaps injustice to use one paper's score to bias the judgement of another, in particular these are two concurrent work submitted to the exact same venue.
>
> (b)	Second, what bothered us sincerely is that this comment severely directed all the attention to this and only this one table that we provided. A good paper is not all about its experimental scores from one sole table (though our FETCH is still better). There are many other factors like novelty, methodological innovation and etc. We wish all the other readers of our paper could also enjoy reading the other parts, rather than focusing on this disputed and misled experiment comparisons posted.
>
> (c)	Third, in our original manuscript, we have fairly compared our methods with the current state-of-the-art and peer-reviewed AutoFE baselines and also submitted our code. We do believe these results are enough to judge our methodological novelty. Though it is known the community does not require comparisons to a work that is not yet peer-reviewed, we still spent a whole day on making the experiments right. The purpose of these results are not to escalate the disputation here. Rather, we simply want to make things right, fair and trustworthy. And we hope our precious time can be spent responding to valuable, fair, and friendly comments.
>
> At last, while we aren't the expert to interpret the anonymity rules. We are not sure whether making comments naming a concurrent work from the same venue has or hasn't violated the rules. Again, we do not want to see this to escalate. To protect you and your paper, as well as protecting ours, we will stop the discussion of this thread right here --- perhaps except for posting the results that we find out in the future.

---

### Author Response · Authors · 2022-11-15
**Revision Update 1 of Supplementary Material**

**About the Update:**

To better help people reproduce the experiment, we have reorganized and updated the code in our supplementary material. We have also added numerous comments to the code for a clearer understanding. The dependencies required to run the code are listed in ``requirement.txt``.

The updated paper also is modified as reviewers suggested.

---

### Public Comment · ~Mark_Wang1 · 2022-11-19
**This is an updated version of our previous response.**

Dear FETCH authors.

We agree that it would be perfectly fine that a paper does not compare with concurrent submission. In this regard, our first comments may unfortunately bring unwanted attention that could negatively impact the reviewers' judgement. It was a bit late for us to published further response since ICLR didn't allow public comments for some time.  Fortunately, it turned out that this does not impact the reviewers' and AC's decision.

While we still think it is important to report out-of-sample scores, we will leave a formal discussion and comparison to the next version of the OpenFE paper. Indeed, many experimental details may impact the performance, and it would be hasty to publish comparison results during the short period of the rebuttal. Hence, we retract the experimental results in our second response, and we will make a formal experimental comparison in the OpenFE paper.

Finally, our previous response did not mean to be unfriendly, and we hope our discussion is only about machine learning.
Indeed, we really appreciate that FETCH authors care this problem as we do and provide interesting methods for tackling it.

Thanks for your attention.

OpenFE authors.

---

### Decision · Program_Chairs · 2023-01-20

**Decision:**

Accept: notable-top-25%

**Justification For Why Not Higher Score:**

No theoretical analysis is provided regarding the transferability of the proposed automated feature engineering framework.

**Justification For Why Not Lower Score:**

The idea is interesting. The empirical results are promising, especially for the transfer learning setting.

**Metareview: Summary, Strengths And Weaknesses:**

In this paper, the authors proposed a new automated feature engineering framework. The proposed framework can be transferred across different datasets, which is interesting. Empirical results are promising. One potential limitation is that the proposed framework may not be applied to different application domains beyond tabular datasets. Though the authors added some discussions about how to extend the framework to non-tabular data, they did not provide empirical evidence that the proposed framework can perform well on non-tabular data.

In summary, this is a good paper for ICLR-23.

**Note From Pc:**

if the above contains the word "oral" or "spotlight" please see: "oral" presentation means -> notable-top-5% and "spotlight" means -> notable-top-25%. As stated in our emails, we are disassociating presentation type from AC recommendations